# Understanding and Leveraging Overparameterization in Recursive Value Estimation

**Chenjun Xiao**[1,2,*]**, Bo Dai**[1]**, Jincheng Mei**[1]**, Oscar Ramirez**[1]**, Ramki Gummadi**[1]**,
Chris Harris**[1] **& Dale Schuurmans**[1,2]
[1]Google
[2]Department of Computing Science, University of Alberta

## Abstract

The theory of function approximation in reinforcement learning (RL) typically considers low capacity representations that incur a tradeoff between approximation error, stability and generalization. Current deep architectures, however, operate in an overparameterized regime where approximation error is not necessarily a bottleneck. To better understand the utility of deep models in RL we present an analysis of recursive value estimation using *overparameterized* linear representations that provides useful, transferable findings. First, we show that classical updates such as temporal difference (TD) learning or fitted-value-iteration (FVI) converge to *different* fixed points than residual minimization (RM) in the overparameterized linear case. We then develop a unified interpretation of overparameterized linear value estimation as minimizing the Euclidean norm of the weights subject to alternative constraints. A practical consequence is that RM can be modified by a simple alteration of the backup targets to obtain the same fixed points as FVI and TD (when they converge), while universally ensuring stability. Further, we provide an analysis of the generalization error of these methods, demonstrating per iterate bounds on the value prediction error of FVI, and fixed point bounds for TD and RM. Given this understanding, we then develop new algorithmic tools for improving recursive value estimation with deep models. In particular, we extract two regularizers that penalize out-of-span top-layer weights and co-linearity in top-layer features respectively. Empirically we find that these regularizers dramatically improve the stability of TD and FVI, while allowing RM to match and even sometimes surpass their generalization performance with assured stability.

## 1 Introduction

Model-free value estimation remains a core method of reinforcement learning (RL), lying at the heart of some of the most prominent achievements in this area (Mnih et al., 2015; Bellemare et al., 2020). Such success appears paradoxical however, given that value estimation is subject to the *deadly triad*: any value update that combines off-policy estimation with Bellman-bootstrapping and function approximation diverges in the worst case (Sutton and Barto, 2018). Without additional assumptions it is impossible to ensure the viability of iterative value estimation schemes, yet this remains a dominant method in RL—its popularity supported by empirical success in many applications. Such a sizable gap between theory and practice reflects limited understanding of such methods, how they behave in practice, and what accounts for their empirical success (van Hasselt et al., 2018; Achiam et al., 2019).

Decomposing the deadly triad indicates that off-policy estimation and bootstrapping are difficult to forego: off-policy estimation is supported by the empirical effectiveness of action value maximization and replay buffers, while Bellman-bootstrapping provides significant advantages over Monte Carlo estimation (Sutton, 1988). On the other hand, our understanding of the third factor, the relationship between function representation and generalization, has evolved dramatically in recent years. Although it was once thought that *restrictive* function approximation—representations that lack capacity to fit all data constraints—might be essential for generalization, we now know that this view is oversimplified (Belkin et al., 2019). The empirical success of deep learning (Krizhevsky et al., 2012),

---

*Work performed while an intern at Google Brain. Email: {chenjun,daes}@ualberta.ca

extremely large models (Brown et al., 2020) and associated theoretical advances (Jacot et al., 2018) have made it clear that gradient-based training of overparameterized models embodies implicit biases that encourage generalization even after all data constraints are fit exactly. This success suggests a new opportunity for breaking the deadly triad: by leveraging overparameterized value representations one can avoid some of the most difficult tradeoffs in value-based RL (Lu et al., 2018).

The use of overparameterized deep models in value-based RL, however, still exhibits mysteries in stability and performance. Although one might expect larger capacity models to improve the stability of Bellman-bootstrapping, in fact the opposite appears to occur (van Hasselt et al., 2018). Our own empirical experience indicates that classical value estimation with deep models eventually diverges in non-toy problems. It has also been shown that value updating leads to premature rank-collapse in deep models (Kumar et al., 2021), coinciding with instability and degrading generalization. In practice, some form of *early-stopping* is usually necessary to obtaining successful results, a fact that is not often emphasized in the literature (Agarwal et al., 2021). Meanwhile, there is a long history of convergent methods being proposed in the RL literature—starting from residual gradient (Baird, 1995), to gradient-TD (Sutton et al., 2008; Maei et al., 2009), prox gradient TD (Liu et al., 2015; 2016), and emphatic TD (Yu, 2015; Sutton et al., 2016)—yet none of these has demonstrated sufficient generalization quality to supplant unstable methods. The current state of development leaves an awkward tradeoff between stability and generalization. A stable recursive value estimation method that ensures generalization quality with overparametrization remains elusive.

In this paper we investigate whether overparameterized value representations might allow the stability-generalization tradeoff to be better managed, enabling stable estimation methods that break the deadly triad and generalize well. We first consider policy evaluation with *overparameterized linear* value representations, a simplified setting that still imposes the deadly triad (Zhang et al., 2021). Here we find that alternative updates, such as temporal differencing (TD), fitted value iteration (FVI) and residual minimization (RM) converge to *different* fixed points in the overparameterized case (when they converge), even though these updates share a common fixed point when the approximation error is zero and there are no extra degrees of freedom (Dann et al., 2014). That is, these algorithms embody implicit biases that *only* become distinguishable in the overparameterized regime. From this result, we observe that the fixed points lie in different bases, which we use to develop a unified view of iterative value estimation as minimizing the Euclidean norm of the weights subject to alternative constraint sets. This unification allows us to formulate alternative updates that share fixed points with TD and FVI but guarantee stability without requiring regularization or prox constraints (Zhang et al., 2021). Next, we analyze the generalization performance of these algorithms and provide a per-iterate bound on the value estimation error of FVI, and fixed point bounds on the value estimation error of TD. From these results, we identify two novel regularizers, one that closes the gap between RM and TD and another that quantifies the effect of the feature representation on the generalization bound. We deploy these regularizers in a realistic study of deep model training for optimal value estimation and observe systematic stability and generalization improvements. We also observe that the performance gap between RM and TD can be closed and in some cases eliminated.

## 2 RELATED WORK

Value estimation has a lengthy history throughout RL research. Our main focus is on off-policy value estimation with parametric function representations and iterative (i.e., gradient based) updates. We do not consider exploration nor full planning problems (i.e., approximately solving an entire Markov decision process (MDP)) in the theoretical development, but instead focus on offline value estimation; however, we do apply the findings to policy improvement experiments in the empirical investigation.

Dann et al. (2014) provide a comprehensive survey of value estimation with parametric function representations. Significant attention has been focused on *under*parameterized representations where backed up values are not necessarily expressible in the function class, however we focus on the overparameterized case where any backed up values can be assumed to be exactly representable with respect to finite data. This change fundamentally alters the conclusions one can draw about algorithm behavior, as we see below. One of the key consequences is that classical distinctions (Scherrer, 2010; Dann et al., 2014) between objectives—e.g., mean squared Bellman error (MSBE), mean squared *projected* Bellman error (MSPBE), mean squared temporal difference error (MSTDE), and the norm of the expected TD update (NEU)—all collapse when the Bellman errors can all be driven to zero. Despite this collapse, we find that algorithms targeting the different objectives—TD and LSTD for MSPBE (Sutton, 1988; Bradtke and Barto, 1996) and RM without double sampling (DS) for MSTDE

(Maei et al., 2009; Dann et al., 2014)—converge to different fixed points given overparameterization, even when they ultimately satisfy the same set of temporal consistency constraints.

It is well known that classical value updates can diverge given off-policy data and parametric function representations (Baird, 1995; Tsitsiklis and Van Roy, 1996; 1997). The stability of these methods has therefore been studied extensively with many mitigations proposed, including restricting the function representation (Gordon, 1995; Szepesvári and Smart, 2004) or adjusting the representation to ensure contraction (Kolter, 2011; Ghosh and Bellemare, 2020; Wang et al., 2021b), or modifying the updates to achieve convergent variations, such as LSTD (Bradtke and Barto, 1996; Yu, 2010), FVI (Ernst et al., 2005; Munos and Szepesvári, 2005; Szepesvári and Munos, 2008; Lizotte, 2011) or the introduction of target networks (Mnih et al., 2015; Lillicrap et al., 2016; Zhang et al., 2021; Carvalho et al., 2020). Others have considered modified the updates to combat various statistical inefficiencies (van Hasselt, 2010; Weng et al., 2020; Konidaris et al., 2011). Another long running trend has been to consider two time-scale algorithms and analyses, reflected in gradient-TD methods (Sutton et al., 2008; Maei et al., 2009), prox gradient TD (Liu et al., 2015; 2016), primal-dual TD (Dai et al., 2017; Du et al., 2017), and emphatic TD (Yu, 2015; Sutton et al., 2016). Beyond mere convergence, however, we discover a greater diversity in fixed points among algorithms in the overparameterized case, which play a critical but previously unacknowledged role in generalization quality.

The fact that minimizing MSPBE via TD methods still dominates practice appears surprising given the theoretical superiority of other objectives. It has been argued, for example, that direct policy gradient methods (Sutton et al., 1999) dominate minimizing Bellman error objectives (Geist et al., 2017). Even among Bellman based approaches, it is known that MSBE can upper bound the value estimation error (MSE) whereas MSPBE cannot (Kolter, 2011; Dann et al., 2014), yet MSPBE minimization (via TD based methods) empirically dominates minimizing MSBE (via residual methods). This dominance has been thought to be due to the double sampling bias of residual methods (Baird, 1995; Dann et al., 2014), but we uncover a more interesting finding that their fixed points lie in different bases in the overparameterized setting, and that reducing this difference closes the performance gap.

We analyze the convergence of classical updates given offline data and provide associated generalization bounds, with the primary goal of understanding the discrepancy between previous theory and the empirical success of TD/FVI versus RM. Although this theory sheds new light in exploitable ways, it cannot overcome theoretical limits on offline value estimation, such as lower bounds on worst case error that are exponential in horizon length (Wang et al., 2021a;b; Zanette, 2021; Xiao et al., 2021). We analyze the convergence of the expected updates, extendible to the stochastic case using known techniques (Yu, 2010; Bhandari et al., 2018; Dalal et al., 2018; Prashanth et al., 2021; Patil et al., 2021). We expand the coverage of these earlier works by including alternative updates and focusing on the overparameterized case, uncovering previously unobserved differences in the fixed points.

There is a growing body of work on linear value estimation and planning that leverages the insight of (Parr et al., 2008; Taylor and Parr, 2009) that linear value estimation is equivalent to linear model approximation. A number of works have strived to obtain provably efficient algorithms for approximating the optimal policy values in this setting, but these generally rely on exploration or strong assumptions about data coverage (Song et al., 2016; Yang and Wang, 2019; Duan et al., 2020; Agarwal et al., 2020; Jin et al., 2020; Yang et al., 2020; Hao et al., 2021) that we do not make. Instead we study linear value estimation to gain insight, but rather than focus on linear planning we leverage the findings to improve the empirical performance of value estimation with deep models.

## 3 PRELIMINARIES

**Notation**  We let $\mathbb{R}$ denote the set of real numbers, $\boldsymbol{I}_n$ an $n \times n$ identity matrix, and $\mathbb{I}$ the indicator function. For a finite set $\mathcal{X}$, we use $\Delta(\mathcal{X})$ to denote the set of probability distributions over $\mathcal{X}$. For a vector $\boldsymbol{\mu}$ we let $|\text{supp}(\boldsymbol{\mu})|$ denote the size of the support of $\boldsymbol{\mu}$ (i.e., the number of nonzero entries in $\boldsymbol{\mu}$). For a matrix $\boldsymbol{A} \in \mathbb{R}^{n \times m}$, we let $\boldsymbol{A}^{\dagger}$ be the Moore-Penrose pseudoinverse of $\boldsymbol{A}$, $\|\boldsymbol{A}\|$ be its spectral norm, and $\lambda_{\max}(\boldsymbol{A})$ and $\lambda_{\min}(\boldsymbol{A})$ be its maximum and minimum *non-zero* eigenvalues. We also use $\Pi_{\boldsymbol{A}} = \boldsymbol{A}^{\dagger}\boldsymbol{A}$ to denote the projection matrix to the row space of $\boldsymbol{A}$. For a vector $\boldsymbol{x} \in \mathbb{R}^d$, we let $\|\boldsymbol{x}\|$ be its $l_2$ norm and $\|\boldsymbol{x}\|_{\boldsymbol{A}} = \sqrt{\boldsymbol{x}^{\top}\boldsymbol{A}\boldsymbol{x}}$ be the associated norm for a positive definite matrix $\boldsymbol{A}$. We also use $\text{diag}(\boldsymbol{x}) \in \mathbb{R}^{d \times d}$ to denote a diagonal matrix whose diagonal elements are $\boldsymbol{x}$.

**Markov reward processes**  We consider the problem of predicting the value of a given stationary policy in a Markov Decision Process (MDP). For a stationary policy, this problem can be formulated

in terms of a Markov reward process $M = \{\mathcal{S}, P, r, \gamma\}$, such that $\mathcal{S}$ is a finite set of states, $r : \mathcal{S} \to \mathbb{R}$ and $P : \mathcal{S} \to \Delta(\mathcal{S})$ are the reward and transition functions respectively, and $\gamma \in [0, 1)$ is the discount factor. Let $S = |\mathcal{S}|$ be the number of states. For a given state $s \in \mathcal{S}$, the function $r(s)$ gives the immediate reward incurred at $s$, while $P(\cdot|s)$ gives the next-state transition probability of $s$. The value function specifies the future discounted total reward obtained from each state, defined as

$$v(s) = \mathbb{E}\left[\sum_{t=0}^{\infty} \gamma^t r(s_t) \Big| s_0 = s\right]. \tag{1}$$

To simplify the presentation we identify functions as vectors to allow vector-space operations: the value function $v$ and reward function $r$ are identified as vectors $v, r \in \mathbb{R}^S$, the transition $P$ is identified as an $S \times S$ transition matrix, where the $s$-th row $P_s$ specifies the transition probability $P(\cdot|s)$ of state $s$. These definitions allow the value function to be expressed using Bellman's equation

$$v = r + \gamma P v. \tag{2}$$

**Linear Function Approximation**  It is usually not possible to consider tabular value representations in practice, since the state set is usually combinatorial or infinite. In our theoretical development we focus on linear function approximations, where $v$ is approximated by a linear combination of features describing states; i.e., $v(s) \approx \phi(s)^\top \boldsymbol{\theta}$, where $\boldsymbol{\theta} \in \mathbb{R}^d$ is a parameter vector and $\phi : \mathcal{S} \to \mathbb{R}^d$ maps a given state $s \in \mathcal{S}$ to a $d$-dimensional feature vector $\phi(s) \in \mathbb{R}^d$. We let $\boldsymbol{\Phi} \in \mathbb{R}^{|\mathcal{S}| \times d}$ denote the feature matrix, with the $s$-th row corresponding to the feature vector $\phi(s)$, so that the value approximation can be written as $v \approx \boldsymbol{\Phi\theta}$. We assume $\|\phi(s)\| \leq 1$ for any $s \in \mathcal{S}$, and for simplicity we also assume that there is no redundant or irrelevant features in the feature map; that is, $\boldsymbol{\Phi}$ is full rank.

### 3.1 BATCH VALUE ESTIMATION

We consider *batch mode* ("offline") estimation of the value function. Let $\boldsymbol{\mu} \in \Delta(\mathcal{S})$ be an arbitrary probability distribution over states and $\boldsymbol{D_\mu} = \text{diag}(\boldsymbol{\mu})$. The data set consists of $\{s_i, r_i, s_i'\}_{i=1}^n$ transition tuples, which are generated by $s \sim \boldsymbol{\mu}, r_i = r(s_i), s_i' \sim P(\cdot|s_i)$. Let $n(s) = \sum_{i=1}^n \mathbb{I}(s_i = s)$ be the number of counts of state $s$. We define the *empirical data distribution matrix* $\boldsymbol{D} = \text{diag}(\hat{\boldsymbol{\mu}})$, where $\hat{\mu}(s) = n(s)/n$ is the empirical data distribution over states. The goal is to estimate the value function by finding a weight vector $\boldsymbol{\theta} \in \mathbb{R}^d$ that minimizes the *value prediction error*,

$$\mathcal{E}(\boldsymbol{\theta}) = \|\boldsymbol{\Phi\theta} - v\|_{\boldsymbol{D_\mu}}^2 = \sum_{s \in \mathcal{S}} \mu(s)(\phi(s)^\top \boldsymbol{\theta} - v(s))^2. \tag{3}$$

Let $\hat{\boldsymbol{P}}$ be the empirical transition matrix, where the $s$-th row represents the estimated transition of state $s$: if $n(s) > 0$, $\hat{\boldsymbol{P}}_s(s') = \sum_{i=1}^n \mathbb{I}(s_i = s, s_i' = s')/n(s)$; if $n(s) = 0$, $\hat{\boldsymbol{P}}_s(s') = 0$ for all $s' \in \mathcal{S}$. The empirical mean squared *Bellman error* on the batch data can be defined as

$$\text{MSBE}(\boldsymbol{\theta}) = \tfrac{1}{2}\big\|r + \gamma \hat{\boldsymbol{P}}\boldsymbol{\Phi\theta} - \boldsymbol{\Phi\theta}\big\|_{\boldsymbol{D}}^2. \tag{4}$$

**Over vs Underparameterized Features**  In this paper we are particularly interested in the *overparameterized* regime $d > |\text{supp}(\hat{\boldsymbol{\mu}})|$ where one can exactly satisfy the temporal consistencies on all transitions in the batch data set, achieving zero Bellman error. (Obviously this would also be possible if $d = |\text{supp}(\hat{\boldsymbol{\mu}})|$ but the strictly overparameterized case is more interesting, as we will see below.) By contrast, in the underparameterized regime $d < |\text{supp}(\hat{\boldsymbol{\mu}})|$, one can only expect to find an approximate solution that in general has nonzero Bellman error.

We consider three core algorithms in our analysis, covering major classical approaches.

**Residual Minimization (RM)**  RM directly minimizes the empirical mean squared Bellman error Eq. (4) (MSBE) (Baird, 1995). The gradient update (Dann et al., 2014) can be expressed as

$$\boldsymbol{\theta}_{t+1} = \boldsymbol{\theta}_t - \eta(\boldsymbol{\Phi} - \gamma\hat{\boldsymbol{P}}\boldsymbol{\Phi})^\top \boldsymbol{D}\big(\boldsymbol{\Phi\theta}_t - (r + \gamma\hat{\boldsymbol{P}}\boldsymbol{\Phi\theta}_t)\big), \tag{5}$$

where $\boldsymbol{\theta}_t$ is the estimated weight at step $t$, and $\eta$ is the learning rate. As a gradient descent method, the convergence of this update is robust, and applies to both linear and nonlinear function approximation.

**Temporal Difference (TD) Learning**  The simplest variant of TD (Sutton, 1988), known as TD(0), also updates weights iteratively using transition data to approximate the value function. Let $\boldsymbol{\theta}_t$ be the weight vector at step $t$. Then the so-called "*semi-gradient*" of Eq. (4) is used to compute the update,

$$\boldsymbol{\theta}_{t+1} = \boldsymbol{\theta}_t - \eta\boldsymbol{\Phi}^\top \boldsymbol{D}\big(\boldsymbol{\Phi\theta}_t - \big(r + \gamma\hat{\boldsymbol{P}}\boldsymbol{\Phi\theta}_t\big)\big), \tag{6}$$

where $\eta$ is the learning rate. From Eq. (6), it is clear that in the underparameterized ($d < |\text{supp}(\hat{\boldsymbol{\mu}})|$) regime, if the system converges, it must converge to parameters $\boldsymbol{\theta}_D^*$ such that

$$\boldsymbol{\Phi}^\top \boldsymbol{D} r - \boldsymbol{\Phi}^\top \boldsymbol{D}(\boldsymbol{\Phi} - \gamma \hat{\boldsymbol{P}} \boldsymbol{\Phi})\boldsymbol{\theta}_D^* = 0 \quad \Rightarrow \quad \boldsymbol{\theta}_D^* = (\boldsymbol{\Phi}^\top \boldsymbol{D}(\boldsymbol{\Phi} - \gamma \hat{\boldsymbol{P}} \boldsymbol{\Phi}))^{-1} \boldsymbol{\Phi}^\top \boldsymbol{D} r \,, \quad (7)$$

where $\boldsymbol{\theta}_D^*$ is the *TD fixed point*. That is, given limited representational power, the TD fixed point minimizes the squared projected Bellman error (MSPBE) by solving the *projected Bellman equation*:

$$\boldsymbol{\Phi}\boldsymbol{\theta}_D^* = \Pi_{\boldsymbol{\Phi}}^D\left(r + \gamma \hat{\boldsymbol{P}} \boldsymbol{\Phi} \boldsymbol{\theta}_D^*\right), \quad (8)$$

such that $\Pi_{\boldsymbol{\Phi}}^D = \boldsymbol{\Phi}(\boldsymbol{\Phi}^\top \boldsymbol{D} \boldsymbol{\Phi})^{-1}\boldsymbol{\Phi}^\top \boldsymbol{D}$ is a weighted projection matrix. It is well-known that TD(0) can diverge if the data sampling distribution $\mu$ is not the stationary distribution of the Markov process. One can still compute the TD fixed point directly using batch data, for example using the LSTD algorithm (Bradtke and Barto, 1996), but this requires computation on the order of $O(d^2)$ compared to $O(d)$ of the iterative update algorithm Eq. (6). The value prediction error of TD is discussed in (Tsitsiklis and Van Roy, 1997; Kolter, 2011; Dann et al., 2014; Bhandari et al., 2018).

**Fitted Value Iteration (FVI)**  FVI iteratively updates the weight vector by solving a regression problem where the target is constructed from the current estimate (Ernst et al., 2005; Dann et al., 2014), which is also known as approximate dynamic programming (Sutton and Barto, 2018). In particular, given the current weight $\boldsymbol{\theta}_t$ at iteration $t$, the objective Eq. (9) is minimized to obtain $\boldsymbol{\theta}_{t+1}$,

$$\text{FVI}_t(\boldsymbol{\theta}) = \tfrac{1}{2}\left\|r + \gamma \hat{\boldsymbol{P}} \boldsymbol{\Phi} \boldsymbol{\theta}_t - \boldsymbol{\Phi}\boldsymbol{\theta}\right\|_D^2. \quad (9)$$

A simple calculation shows the TD fixed point matches the fixed point of FVI whenever $\boldsymbol{\theta}_0$ is in the row-span of $\boldsymbol{D}\boldsymbol{\Phi}$. Although convergence of FVI can be established under strong conditions (Szepesvári and Munos, 2008), the algorithm can be quite unstable in the general batch setting (Chen and Jiang, 2019; Wang et al., 2021b).

## 4  OVER-PARAMETERIZED LINEAR VALUE FUNCTION APPROXIMATION

In this section, we study the convergence properties of the value estimation algorithms introduced in Section 3.1 in the *overparameterized* regime where $d > |\text{supp}(\hat{\boldsymbol{\mu}})|$. To faciliate analysis, we first introduce additional notation to simplify the derivations. Let $\{x_i\}_{i=1}^k$ denote the states in the support of $\hat{\boldsymbol{\mu}}$, such that $n(x_i) > 0$ for all $i = \{1, \ldots, k\}$ and $k = |\text{supp}(\hat{\boldsymbol{\mu}})|$. Define a *mask matrix* $\boldsymbol{H} \in \mathbb{R}^{k \times |\mathcal{S}|}$ and a *truncated empirical data distribution matrix* $\boldsymbol{D}_k \in \mathbb{R}^{k \times k}$ according to

$$\boldsymbol{H} = \begin{bmatrix} \mathbf{1}_{x_1}^\top \\ \vdots \\ \mathbf{1}_{x_k}^\top \end{bmatrix}, \qquad \boldsymbol{D}_k = \begin{bmatrix} \hat{\boldsymbol{\mu}}(x_1) & & \\ & \ddots & \\ & & \hat{\boldsymbol{\mu}}(x_k) \end{bmatrix}, \quad (10)$$

where $\mathbf{1}_{x_i} \in \{0, 1\}^{|\mathcal{S}|}$ is an indicator vector such that $\phi(x_i) = \boldsymbol{\Phi}^\top \mathbf{1}_{x_i}$. We can then translate between the full distribution and its support via the following.

**Proposition 1.** *The empirical data distribution matrix $\boldsymbol{D}$ can be decomposed as $\boldsymbol{D} = \boldsymbol{H}^\top \boldsymbol{D}_k \boldsymbol{H}$.*

Let $\boldsymbol{M} = \boldsymbol{H}\boldsymbol{\Phi}$, $\boldsymbol{N} = \boldsymbol{H}\hat{\boldsymbol{P}}\boldsymbol{\Phi}$ and $\boldsymbol{R} = \boldsymbol{H}r$ denote the state features, the expected next state features under the empirical transitions, and the rewards on the support of the data distribution respectively.

**Overparameterized Residual Minimization**  We first study the convergence of RM given a fixed $\boldsymbol{D}$. First note that the update Eq. (5) can be re-written as

$$\boldsymbol{\theta}_{t+1} = (\boldsymbol{I}_d - \eta(\boldsymbol{M} - \gamma\boldsymbol{N})^\top \boldsymbol{D}_k(\boldsymbol{M} - \gamma\boldsymbol{N}))\boldsymbol{\theta}_t + \eta(\boldsymbol{M} - \gamma\boldsymbol{N})^\top \boldsymbol{D}_k \boldsymbol{R}. \quad (11)$$

In the overparameterized regime, one can easily verify that there are infinitely many solutions $\boldsymbol{\theta} \in \mathbb{R}^d$ satisfying $(\boldsymbol{M} - \gamma\boldsymbol{N})\boldsymbol{\theta} = \boldsymbol{R}$. The gradient of Eq. (11) is zero at any of these solutions, which implies that RM can have infinitely many fixed points. However, given that RM minimizes the MSBE objective via gradient descent, as we show in the following theorem, the RM update initialized from $\boldsymbol{\theta}_0 = 0$ will converge to a unique fixed point.

**Theorem 1.** *With $\eta \leq \frac{1}{(1+\gamma)^2}$ and starting from $\boldsymbol{\theta}_0 = 0$, RM converges to $\boldsymbol{\theta}_{RM} = (\boldsymbol{M} - \gamma\boldsymbol{N})^\dagger \boldsymbol{R}$.*

**Remark 1.** *For simplicity we present the fixed points of RM and TD starting from $\boldsymbol{\theta}_0 = 0$. The fixed points given an arbitrary initial weight vector $\boldsymbol{\theta}_0 \in \mathbb{R}^d$ are shown in Appendices A.1 and A.2.*

This result parallels similar findings in the supervised learning literature, that training overparameterized deep models with gradient descent (or related algorithms) encodes implicit regularization

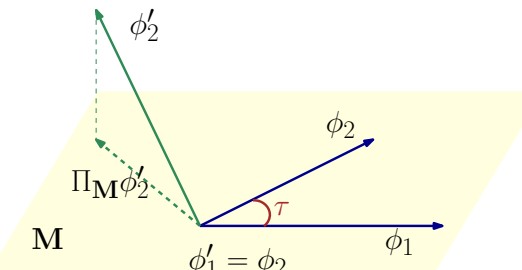

Figure 1: *An illustrative example showing the spectrum of $W$ with $k = 2$ and $d = 3$. $M = [\phi_1, \phi_2]^\top$. Without loss of generality, let $\phi_1 = (\cos\tau, \sin\tau, 0)$ and $\phi_2 = (1, 0, 0)$. $N = [\phi_1', \phi_2']^\top$, where $\phi_1' = \phi_2$, $\phi_2' = (-\cos\tau, \frac{\sqrt{2}}{2}\sin\tau, \frac{\sqrt{2}}{2}\sin\tau)$. Then $W = [[0, 1]^\top, [\frac{\sqrt{2}}{2}, -(1 + \frac{\sqrt{2}}{2}\cos\tau)]^\top]$. Clearly, the spectral norm of $W$ increases as the angle $\tau$ between $\phi_1$ and $\phi_2$ decreases.*

that drives the model solution to particular outcomes in the overparameterized regime (Soudry et al., 2018; Gunasekar et al., 2018; Neyshabur et al., 2019). Moreover, this implicit regularization is often associated with generalization benefits. However, unlike the case for supervised learning, RM solutions do not often generalize well. Below we uncover a key difference between the RM fixed point and those of TD and FVI that sheds new light on the source of generalization differences.

**Overparameterized TD Learning**  We next consider the convergence properties of the TD(0) update in the overparameterized setting. First, rewrite the TD(0) update formula Eq. (6) as

$$\theta_{t+1} = (I_d - \eta M^\top D_k(M - \gamma N))\theta_t + \eta M^\top D_k R. \tag{12}$$

Similar to RM, in the overparameterized regime any solutions $\theta \in \mathbb{R}^d$ that satisfy $(M - \gamma N)\theta = R$ are the fixed points of Eq. (12), which implies an infinite set of fixed points. This is quite unlike the underparameterized case where there is a unique TD fixed point Eq. (7) given by the solution of projected Bellman equation. However, we now show that in the overparameterized setting, similar to solving RM using gradient descent, TD also encodes an implicit bias toward a particular fixed point.

This of course requires TD to converge, which can be assured by a simple condition. Let $W = NM^\dagger$, which has a geometric interpretation that we will exploit later in Section 5. Observe that

$$N = N\Pi_M + N(I_d - \Pi_M) = NM^\dagger M + N(I_d - \Pi_M) = WM + N(I_d - \Pi_M), \tag{13}$$

i.e., $N$ can be decomposed into its projection onto the row-span of $M$ plus a perpendicular component. Eq. (13) shows that $W$ projects $N$ onto the row space of $M$; see Fig. 1 for an illustration. We refer to $W$ as the *core matrix* since its norm determines the convergence of TD.

**Theorem 2.** *Choosing $\eta < \frac{1}{(1+\gamma)\|\Phi\|}$ and starting from $\theta_0 = 0$, if $\|W\| < \frac{1}{\gamma}$, TD(0) converges to $\theta_{TD} = M^\dagger(I_k - \gamma W)^{-1}R$. If $\|W\| \geq \frac{1}{\gamma}$ there is an initial $\theta_0$ for which TD(0) does not converge.*

A few key observations. First, note that the RM fixed point in Theorem 1 and the TD fixed point in Theorem 2 are not identical. That is, the different value estimation algorithms continue to demonstrate different preferences for fixed points, but in the overparameterized setting these differences are *implicit* in the algorithms and cannot be captured by the MSBE versus MSPBE objectives, since both are zero for any $\theta$ that satisfies $(M - \gamma N)\theta = R$. Second, *the fixed point of TD lies in different basis than RM*. That is, $\theta_{TD}$ lies in the row space of the state features $M$, whereas $\theta_{RM}$ lies in the row space of the residual features $M - \gamma N$, and these two spaces are not identical in general. We revisit the significance of this difference below, but intuitively, the parameter vector $\theta$ is being trained to predict values rather than temporal differences, and the future test states from which value predictions are made will tend to be closer to the space spanned by $M$ than $M - \gamma N$.

**Overparameterized Fitted Value Iteration**  Finally, we consider the convergence of FVI. Recall that at iteration $t$, FVI solves the least squares problem Eq. (9) to compute the next weight vector. Using the notation established above, the normal equations for this problem can be expressed as $M^\top D_k M\theta = M^\top D_k(R + \gamma N\theta_t)$, but this system cannot be directly used to compute the solution since $M^\top D_k M$ is not invertible. Furthermore, just like RM and TD, any $\theta \in \mathbb{R}^d$ that satisfies $(M - \gamma N)\theta = R$ is a fixed point of FVI. If one solves the least squares problem Eq. (9) using gradient descent, it is known (Bartlett et al., 2021; Soudry et al., 2018) that the optimization converges to the minimum norm solution

$$\theta_{t+1} = M^\dagger(R + \gamma N\theta_t). \tag{14}$$

Interestingly, by choosing this solution, each iteration of FVI corresponds to applying a linear backup on the current value estimate, where the backup operator is defined by the core matrix.

**Definition 1.** *Define the core matrix linear operator $\mathcal{T}_W$ by $\mathcal{T}_W \nu = R + \gamma W \nu$ for any $\nu \in \mathbb{R}^S$.*

Similar results for the case with underparameterized linear model have been discussed in (Parr et al., 2008). Using this operator we can characterize the convergence condition of FVI, reaching the conclusion that whenever $\mathcal{T}_W$ is a non-expansion, FVI converges to the same fixed point as TD.

**Theorem 3.** *Let $\theta_0$ be the initial weight and $\theta_t \in \mathbb{R}^d$ be the output of FVI at iteration $t$. We have*

$$\theta_{t+1} = M^\dagger \mathcal{T}_W^t (R + \gamma N \theta_0).\tag{15}$$

*Furthermore, given that $\|W\| < 1/\gamma$, the algorithm converges to $\theta_{TD} = M^\dagger (I_k - \gamma W)^{-1} R$. If $\|W\| \geq \frac{1}{\gamma}$ there is an initial $\theta_0$ for which FVI does not converge.*

### 4.1 Unified View of Overparameterized Value Estimators

We now show that the convergence points above can be characterized as solutions to related constrained optimization problems, providing a unified perspective on the respective algorithm biases.

**Theorem 4.** *$\theta_{RM}$ is the solution of the following constrained optimization,*

$$\inf_{\theta \in \mathbb{R}^d} \tfrac{1}{2} \|\theta\|^2 \quad s.t. \; M\theta = R + \gamma N \theta, \tag{16}$$

*and $\theta_{TD}$ is the solution of the following constrained optimization,*

$$\inf_{\theta \in \mathbb{R}^d} \tfrac{1}{2} \|\theta\|^2 \quad s.t. \; M\theta = R + \gamma N \theta, \; \mathrm{null}(M)\theta = 0. \tag{17}$$

That is, the convergence points of RM, TD and FVI in the overparameterized case can all be seen as minimizing the Euclidean norm of the weights $\theta$ subject to satisfying the Bellman constraints $M\theta = R + \gamma N \theta$, where TD and FVI implicitly add the additional constraint that $\theta$ must lie in the row span of $M$; moreover, this is the *only* constraint that differentiates TD from RM. From this perspective, the algorithms can all be seen as iterative procedures for solving a particular form of quadratic program, when they converge. Of course, proper constrained optimization techniques would be able to stably compute solutions in scenarios where TD or FVI diverge (Boyd and Vandenberghe, 2004), but a more direct way to ensure convergence is implied by the following corollary.

**Corollary 1.** *$\theta_{TD}$ is also the solution of the following constrained optimization,*

$$\inf_{\theta \in \mathbb{R}^d} \tfrac{1}{2} \|\theta\|^2 \quad s.t. \; M\theta = R + \gamma N \Pi_M \theta. \tag{18}$$

Note that the right hand side of the constraint simply pre-projects next state value predictions onto the row space of $M$ before determining the Bellman backed up value. This allows a novel objective to be formulated whose minimizer recovers the same fixed point as TD,

$$\mathrm{MSCBE}(\theta) = \tfrac{1}{2} \| R + \gamma N \Pi_M \theta - M\theta \|_D^2, \tag{19}$$

which stands for mean squared *corrected* Bellman error. Note that MSCBE is not identical to MSPBE because the projection is applied before not after the Bellman backup. Gradient descent minimization of MSCBE yields the same fixed point as $\theta_{TD}$, which is essentially equivalent to applying RM to corrected target values while ensuring stability. Note also that in the linear case the projection matrix $\Pi_M$ only needs to be precomputed once.

### 4.2 Value Prediction Error Bounds

One can also establish generalization bounds on the value estimation error of these methods in the overparameterized regime. We first provide a finite time analysis of the value prediction error of FVI.

**Theorem 5.** *Let $\hat{\Sigma} = M^\top D_k M$ be the empirical covariance matrix, and $\theta_t$ be the output of FVI starting from $\theta_0$ as defined in Theorem 3. Then for any $\theta^* \in \arg\min_{\theta \in \mathbb{R}^d} \mathcal{E}(\theta)$,*

$$\mathcal{E}(\theta_t) - \mathcal{E}(\theta^*)$$

$$\leq \frac{1}{k\lambda_{\min}(\hat{\Sigma})} \left( \left( \varepsilon^2 + \sigma^2 \right) \left\| \sum_{i=0}^{t-1} (\gamma W)^i \right\|^2 + \left\| (\gamma W)^{t-1} \right\|^2 \|\Phi\|^2 \|\theta_0 - \theta^*\|^2 \right) + \tfrac{1}{2} \|\theta^*\|_{I_d - \Pi_M}^2, \tag{20}$$

*where $\varepsilon = \|N(I_d - \Pi_M)\theta^*\|$ and $\sigma = \|H(\hat{P} - P)v\|$.*

Intuitively, $\varepsilon$ measures the length of next-state features along the direction $\theta^*$, and $\sigma$ is the expected value prediction error under the empirical transition model, which can be bounded using standard

concentration inequalities. The proof of this theorem is given in Appendix A.5. Observe that for any step $t \geq 1$, the output of FVI $\boldsymbol{\theta}_t$ is within the row-span of $\boldsymbol{M}$. This allows one to decompose the prediction error into a component within the row-span, controlled by leveraging the core matrix linear operator $\mathcal{T}_{\boldsymbol{W}}$, and an orthogonal component that can be bounded by $\|\boldsymbol{\theta}^*\|_{\boldsymbol{I}_d - \Pi_{\boldsymbol{M}}}^2$.

Under the convergence conditions of Theorems 2 and 3, we also have the following generalization bound for the value prediction error of $\boldsymbol{\theta}_{\text{TD}}$.

**Corollary 2.** *Suppose that $\|\boldsymbol{W}\| \leq 1$, and the value of any $s \in \mathcal{S}$ is bounded by $v(s) \in [0, v_{\max}]$. For any $\boldsymbol{\theta}^* \in \arg\min_{\boldsymbol{\theta} \in \mathbb{R}^d} \mathcal{E}(\boldsymbol{\theta})$,*

$$\mathbb{E}[\mathcal{E}(\boldsymbol{\theta}_{\text{TD}})] \leq \frac{\gamma \log(|\mathcal{S}|/\delta)}{n_{\min} \mathbb{E}[\lambda_{\min}(\hat{\boldsymbol{\Sigma}})](1-\gamma)^4} + \frac{4\gamma \mathbb{E}[\|\boldsymbol{\theta}^*\|_{\boldsymbol{I}_d - \Pi_{\boldsymbol{M}}}^2]}{\mathbb{E}[\lambda_{\min}(\hat{\boldsymbol{\Sigma}})](1-\gamma)^2} + \delta v_{\max}, \qquad (21)$$

*where $n_{\min} = \min_{s:n(s)>0} n(s)$ is the minimum counts given the data set.*

This result automatically implies the requirements for ensuring offline generalization, accounting both for distribution shift (Wang et al., 2021b) and policy completeness (Munos and Szepesvári, 2005; Duan et al., 2020) in feature space. In particular, for Eq. (20) and Eq. (21), we characterize the distribution shift using well known concentration bounds in Appendix A.6, which leads to the denominators $k\lambda_{\min}(\hat{\boldsymbol{\Sigma}})$ and $n_{\min}\mathbb{E}[\lambda_{\min}(\hat{\boldsymbol{\Sigma}})]$ respectively. In addition, we explicitly characterize the misalignment between the features of current states and next states using the core matrix, which can be used to bound misalignment between values, replacing the feature completeness assumption.

We note that if the convergence condition cannot be satisfied, that is when $\|\boldsymbol{W}\| \geq 1/\gamma$, the estimation error could be arbitrarily large. The sources of value estimation error are explicit in Corollary 2. The first term measures the error due to sampling (statistical error), while the second term considers out-of-span components of the optimal weight vector $\boldsymbol{\theta}^*$ with respect to $\boldsymbol{M}$ (approximation error). The smallest eigenvalue of the empirical covariance matrix $\mathbb{E}[\lambda_{\min}(\hat{\boldsymbol{\Sigma}})]$, as well as the length of the orthogonal components $\mathbb{E}[\|\boldsymbol{\theta}^*\|_{\boldsymbol{I}_d - \Pi_{\boldsymbol{M}}}^2]$, can both be controlled using classical techniques for concentration properties of random matrix. In Appendix A.7 we present the exact approach for bounding these two terms. Furthermore, by Corollary 1, one can also apply Corollary 2 to an algorithm that directly optimizes MSCBE. Although a solution of Eq. (18) must exist, its value prediction error can be arbitrarily large given that $\|\boldsymbol{W}\| \geq 1/\gamma$. This also connects to a similar result for the TD fixed point that minimizes MSPBE in the underparameterized regime (Kolter, 2011).

## 5 REGULARIZERS FOR DEEP REINFORCEMENT LEARNING ALGORITHMS

For tractability, the theory in prior sections assumes fixed representations with a linear parameterization on only the final layer parameters of the value function. However, in practice, deep RL algorithms also learn the representations in an end-to-end fashion. Inspired by the linear case, we now identify two novel regularizers that are applicable more generally—one that closes the gap between RM and TD inspired by the unified view of different fixed points, and another that quantifies the effect of feature representation on the generalization bound.

**Two-Part Approximation**  Most deep RL algorithms rely on approximating values with a deep neural network $Q_\omega$ that predicts the future outcome of given state-action pair (Mnih et al., 2015; Kalashnikov et al., 2018; Lillicrap et al., 2016). In practice, $Q_\omega$ is trained by TD learning that minimizes the objective $\sum_{s,a}(r(s,a) + \gamma \bar{Q}_\omega(s,a) - Q_\omega(s,a))$, where $\bar{Q}_\omega(s,a)$ is known as the *target network* to increase the learning stability. We view $Q_\omega$ as a two part-approximation with $\omega = (\phi, \theta)$, where the output of the penultimate layer is referred as the feature mapping $\phi$, the weight of last fully connected layer is referred as $\theta$, and the Q-function is approximated by $Q_\omega(s,a) = \phi(s,a)^\top \theta$. Our goal is to define regularizers on $\phi$ and $\theta$ that can be effectively applied to practical algorithms.

The first regularizer directly takes inspiration from Theorem 4: by restricting the linear weight $\theta$ within the row space of $\boldsymbol{M}$ (now defined by exited $(s,a)$ pairs in the data), RM finds the same fixed point as TD. We implement this idea by penalizing the norm of the perpendicular component of $\theta$,

$$\mathcal{R}_\theta = \|\theta - \Pi_{\boldsymbol{M}}\theta\|, \qquad (22)$$

In practice we compute this regularizer for each minibatch of data. The projection step is computed by a least squares algorithm with an additional $l_2$ regularization for numerical stability.

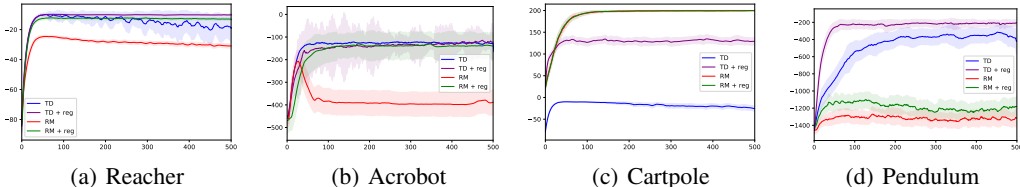

| (a) Reacher | (b) Acrobot | (c) Cartpole | (d) Pendulum |

Figure 2: *We show the results with proposed regularization compared to the baseline algorithms. The algorithms are trained using a fixed offline data set collected by random initialized policies. The $x$-axis shows the training iterations (in thousands) and $y$-axis shows the performance. All plots are averaged over 100 runs. The shaded area shows the standard error.*

The second regularizer is designed to address the effect of the feature representation on convergence and value prediction error. In particular, Theorems 2 and 3 show that TD and FVI converge if the spectral norm of $W$ be upper bounded by $1/\gamma$, which by Theorem 5 will also reduce the bound on generalization error. Hence, it is natural to penalize the norm of this matrix using standard automatic differentiation tools. However, such an approach is prone to numerical difficulty, as it involves differentiation through a matrix pseudo inverse. We instead propose an alternative regularizer inspired by the geometric interpretation of the core matrix Eq. (13): recall from Fig. 1 that $W$ can be viewed as the weights that project $N$ onto the row space of $M$. To ensure that an arbitrary feature vector can be well approximated using $W$, it would be ideal if $M$ was orthonomral, which would imply an ideally-behaved basis to represent $N$. This intuition justifies the following regularization:

$$\mathcal{R}_\phi = \left\| \beta \boldsymbol{I}_d - \boldsymbol{M}^\top \boldsymbol{D}_k \boldsymbol{M} \right\| , \tag{23}$$

where $\beta$ is a scale parameter designed to approximate the column norm. That is, the regularizer forces the neural network to learn an orthogonal feature embedding by normalizing the empirical feature covariance matrix. The gradient of $\mathcal{R}_\phi$ can also approximated using mini-batches. We augment the original learning objectives by adding both $\mathcal{R}_\theta$ and $\mathcal{R}_\phi$ weighted by hyper-parameters.

### 5.1 EMPIRICAL JUSTIFICATION OF REGULARIZERS

The goal of our experiments is to assess the applicability of the proposed regularization schemes based on orthogonality and projection operations to practical deep RL algorithms. To avoid the confounding effects of exploration, we restrict our study to learning from a frozen batch of data with a fixed number of transitions collected prior to learning. We use a *randomly initialized policy* in this initial collection step. We consider both discrete and continuous control benchmarks in this analysis. For the discrete action environments, we use DQN (Mnih et al., 2015) as the baseline algorithm to add our regularizers. For continuous control environments, we use QT-Opt (Kalashnikov et al., 2018) as the baseline algorithm, which is an actor-critic method that applies the cross-entropy method to perform policy optimization. Our modifications add $\mathcal{R}_\phi$ and $\mathcal{R}_\theta$ to the standard MSBE objective on the critic Q-network. Additional details describing the complete experiment setup for each environment are provided in Appendix B. Experimental results contrasting vanilla TD and RM with their regularized variants are summarized in Fig. 2. These findings demonstrate that the proposed regularization schemes can be used to improve the performance of both vanilla TD learning and RM. Note that RM is typically less popular than TD due to its worse empirical performance. On Acrobot and Reacher, the modification was able to fully close the gap between RM and TD. On Cartpole, (where vanilla RM dominates vanilla TD), and on Pendulum, the regularizers also deliver significant improvements to the TD learning baseline and modest improvements to the RM baseline.

## 6 CONCLUSION

We have investigated the fixed points of classical updates for value estimation in the overparameterized setting, where there is sufficient capacity to fit all the Bellman constraints in a given data set. We find that TD and FVI have different fixed points than RM, but in the linear case the difference can be entirely attributed to a constraint missing from RM that the solution lie in the row space of the predecessor state features. We devised two novel regularizers based on these findings, which stabilized the performance of TD without sacrificing generalization, while improving the generalization of RM, in the setting of estimating optimal values with a deep model. Characterizing the implicit bias of other algorithms, such as gradient or emphatic TD variants remains open. Identifying other regularizers that further close the gap between TD and RM is also an interesting direction for future investigation.

## 7 ACKNOWLEDGEMENT

The authors would like to thank Mengjiao Yang, George Tucker, Ofir Nachum and Aviral Kumar for insightful discussions and providing feedback on a draft of this manuscript. Dale Schuurmans gratefully acknowledges funding from the Canada CIFAR AI Chairs Program, Amii and NSERC.

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

# Appendix

## A  PROOFS

### A.1  PROOF OF THEOREM 1

**Theorem 6** (Restatement of Theorem 1). *Let $\theta_0 \in \mathbb{R}^d$ be the initial weight vector. With $\eta \leq \frac{1}{(1+\gamma)^2}$, RM converges to $\theta_{\mathrm{RM}} = (M - \gamma N)^\dagger R + (I_d - \Pi_{M-\gamma N})\theta_0$.*

*Proof.* Let $A = M - \gamma N$ for simplicity. First recall the residual minimization update,

$$\theta_{t+1} = \left(I_d - \eta A^\top D_k A\right)\theta_t + \eta A^\top D_k R. \tag{24}$$

Let $\theta^* = A^\dagger R$. It can be verified $\theta^*$ is one of the feasible solution as $A\theta^* = R$. Then we use induction to show that for any $\theta_0 \in \mathbb{R}^d$ and $t \geq 0$

$$\theta_{t+1} - \theta^* = \left(I_d - \eta A^\top D_k A\right)^{t+1}(\theta_0 - \theta^*). \tag{25}$$

The base case holds by the update rule Eq. (24). Suppose that the statement holds for $t$, then we have

$$\theta_{t+1} - \theta^* = \left(I_d - \eta A^\top D_k A\right)\theta_t + \eta A^\top D_k R - \theta^* \tag{26}$$

$$= \left(I_d - \eta A^\top D_k A\right)\theta_t - (I_d - \eta A^\top D_k A)\theta^* \tag{27}$$

$$= \left(I_d - \eta A^\top D_k A\right)(\theta_t - \theta^*) \tag{28}$$

$$= \left(I_d - \eta A^\top D_k A\right)^{t+1}(\theta_0 - \theta^*). \tag{29}$$

Thus,

$$\theta_{t+1} = (I_d - \eta A^\top D_k A)^{t+1}\theta_0 + (I_d - (I_d - \eta A^\top D_k A)^{t+1})\theta^*. \tag{30}$$

We let $V\Lambda V^\top$ be its eigendecomposition of $A^\top D_k A$, which is the empirical covariance matrix of residual features. Let $V_-$ be the null space of $V$. Then

$$I_d - (I_d - \eta A^\top D_k A)^{t+1} \tag{31}$$

$$= I_d - (VV^\top - \eta V\Lambda V^\top + V_-V_-^\top)^{t+1} \tag{32}$$

$$= I_d - (V(I_k - \eta\Lambda)V^\top + V_-V_-^\top)^{t+1} \tag{33}$$

$$= I_d - (V_-V_-^\top)^{t+1} - V(I_k - \eta\Lambda)^{t+1}V^\top \tag{34}$$

$$= VV^\top - V(I_k - \eta\Lambda)^{t+1}V^\top \tag{35}$$

$$= V\left(I_k - (I_k - \eta\Lambda)^{t+1}\right)V^\top \tag{36}$$

Let $\lambda_{\max}$ be the largest eigenvalue of $A^\top D_k A$. We now show that $\lambda_{\max} \leq 1 + \gamma$.

$$\lambda_{\max}\left(A^\top D_k A\right) \leq \sum_{i=1}^{k} \hat{\mu}(s_i)\lambda_{\max}\left((\phi(s_i) - \gamma\bar{\phi}(s_i'))(\phi(s_i) - \gamma\bar{\phi}(s_i'))^\top\right) \leq (1+\gamma)^2, \tag{37}$$

where we use the fact that $\lambda_{\max}$ is a convex function and we assume $\|\phi(s)\| \leq 1$ for all $s \in \mathcal{S}$. Thus, given that $\eta \leq \frac{1}{1+\gamma}, \eta \leq \frac{1}{\lambda_{\max}}$. Then $I_d - (I_d - \eta A^\top D_k A)^{t+1} = VV^\top$ as $t \to \infty$. Thus

$$\lim_{t\to\infty}\theta_t = \lim_{t\to\infty}(I_d - \eta A^\top D_k A)^{t+1}\theta_0 + VV^\top\theta^* = \lim_{t\to\infty}(I_d - \eta A^\top D_k A)^{t+1}\theta_0 + \theta^*, \tag{38}$$

where the last equality follows by that $\theta^*$ is in the row space of $A$ by definition. When $\theta_0 = 0$, we have the algorithm converges to $\theta^*$.

We next show the result for general $\theta_0$. Let $\theta_0 = \theta_0^1 + \theta_0^2$, where $\theta_0^1 = \Pi_{M-\gamma N}\theta_0$ is the component of $\theta_0$ that is in the row space of $A$, $\theta_0^2 = (I_d - \Pi_{M-\gamma N})\theta_0$ is the perpendicular residual. Then,

$$\lim_{t\to\infty}(I_d - \eta A^\top D_k A)^{t+1}\theta_0 \tag{39}$$

$$= \lim_{t\to\infty}(V_- V_-^\top + V(I_k - \eta\Lambda)^{t+1}V^\top)(\theta_0^1 + \theta_0^2) \tag{40}$$

$$= \theta_0^2 + \lim_{t\to\infty}V(I_k - \eta\Lambda)^{t+1}V^\top\theta_0^1 = \theta_0^2\,, \tag{41}$$

where the last step follows by the choice of $\eta$. This finishes the proof.

$\square$

## A.2 PROOF OF THEOREM 2

We will need the matrix binomial theorem in the proof.

**Lemma 1** (Matrix Binomial Theorem). *For $n \geq 0$ and two matrices $X, Y$*

$$(I + XY)^n X = X(I + YX)^n\,. \tag{42}$$

*Proof.*

$$(I + XY)^n X = \sum_{k=0}^{n}\binom{n}{k}(XY)^k X = X\sum_{k=0}^{n}\binom{n}{k}(YX)^k = X(I + YX)^n\,. \tag{43}$$

$\square$

**Theorem 7** (Restatement of Theorem 2). *Assuming that $M^\top D_k(M - \gamma N)$ is diagonalizable Let $\theta_0 \in \mathbb{R}^d$ be the initial weight vector. With $\eta < \frac{1}{(1+\gamma)\|\Phi\|}$, if $\|W\| < \frac{1}{\gamma}$, TD(0) converges to $\theta_{TD} = M^\dagger(I_k - \gamma W)^{-1}R + \beta$, where $\beta = Q_0 Q_0^{-1}\theta_0$, $Q_0$ are eigenvectors of $M^\top D_k(M - \gamma N)$ with zero eigenvalues. If $\|W\| \geq \frac{1}{\gamma}$ there is an initial $\theta_0$ for which TD(0) does not converge.*

*Proof.* We first rewrite the TD update formulate as

$$\theta_{t+1} = (I_d - \eta M^\top D_k(M - \gamma N))\theta_t + \eta M^\top D_k R \tag{44}$$

A simple recursive argument shows that for any $\theta_0 \in \mathbb{R}^d$,

$$\theta_{t+1} = (I_d - \eta M^\top D_k(M - \gamma N))^{t-1}\theta_0 + \eta\sum_{i=0}^{t}(I_d - \eta M^\top D_k(M - \gamma N))^i M^\top D_k R\,. \tag{45}$$

By the matrix binomial theorem (Lemma 1),

$$(I_d - \eta M^\top D_k(M - \gamma N))^i M^\top D_k = M^\top D_k(I_k - \eta(M - \gamma N)M^\top D_k)^i\,. \tag{46}$$

By writing $N$ as the projection to the row-span of $M$ and the perpendicular component, we have

$$(M - \gamma N)M^\top \tag{47}$$

$$= (M - \gamma N M^\dagger M - \gamma N(I_d - M^\dagger M))M^\top \tag{48}$$

$$= (I_k - \gamma W)MM^\top\,, \tag{49}$$

where the last step follows by $(I_d - M^\dagger M)M^\top = 0$. Thus we can rewrite $\theta_{t+1}$ as

$$\theta_{t+1} = (I_d - \eta M^\top D_k(M - \gamma N))^{t-1}\theta_0 + \eta M^\top D_k\sum_{i=0}^{t}(I_k - \eta(M - \gamma N)M^\top D_k)^i R \tag{50}$$

$$= (I_d - \eta M^\top D_k(M - \gamma N))^{t-1}\theta_0 + \eta M^\top D_k\sum_{i=0}^{t}(I_k - \eta(I_k - \gamma W)MM^\top D_k)^i R\,, \tag{51}$$

Given $\|\boldsymbol{W}\| < 1/\gamma$, we have that all eigenvalues of $\boldsymbol{I}_k - \gamma\boldsymbol{W}$ are positive. Let $\eta < \frac{1}{(1+\gamma)\|\Phi\|}$, then

$$\|\eta(\boldsymbol{I}_k - \gamma\boldsymbol{W})\boldsymbol{M}\boldsymbol{M}^\top\boldsymbol{D}_k\| < \eta\|(\boldsymbol{I}_k - \gamma\boldsymbol{W})\|\|\boldsymbol{M}\boldsymbol{M}^\top\boldsymbol{D}_k\| < 1, \tag{52}$$

otherwise the matrix power series diverges. Thus

$$\eta\boldsymbol{M}^\top\boldsymbol{D}_k\sum_{i=0}^{t}(\boldsymbol{I}_k - \eta(\boldsymbol{I}_k - \gamma\boldsymbol{W})\boldsymbol{M}\boldsymbol{M}^\top\boldsymbol{D}_k)^i\boldsymbol{R} = \boldsymbol{M}^\dagger(\boldsymbol{I}_k - \gamma\boldsymbol{W})^{-1}\boldsymbol{R}. \tag{53}$$

Therefore, given that $\theta_0 = 0$, we have the algorithm converge to $\boldsymbol{M}^\dagger(\boldsymbol{I}_k - \gamma\boldsymbol{W})^{-1}\boldsymbol{R}$.

We now show the convergence point for an arbitrary $\theta_0$. Let $\boldsymbol{Q}\Lambda\boldsymbol{Q}^{-1}$ be the eigen decomposition of $\boldsymbol{M}^\top\boldsymbol{D}_k(\boldsymbol{M} - \gamma\boldsymbol{N})$. By the low rank structure of this matrix, it has at most $h \leq k$ non-zero eigenvalues. Let $\boldsymbol{Q}_0$ be the eigenvectors with eigenvalue *zero*. Then

$$\lim_{t\to\infty}(\boldsymbol{I}_d - \eta\boldsymbol{M}^\top\boldsymbol{D}_k(\boldsymbol{M} - \gamma\boldsymbol{N}))^t\theta_0 \tag{54}$$

$$= \lim_{t\to\infty}\boldsymbol{Q}(\boldsymbol{I}_d - \eta\Lambda)^t\boldsymbol{Q}^{-1}\theta_0 \tag{55}$$

$$= \boldsymbol{Q}_0\boldsymbol{Q}_0^{-1}\theta_0, \tag{56}$$

where the last step follows by the choice of $\eta$.

### A.2.1 Characterization for Non-diagonalizable Case

In the above analysis, we assume that the matrix $\boldsymbol{M}^\top\boldsymbol{D}_k(\boldsymbol{M} - \gamma\boldsymbol{N})$ is diagonalizable. We now characterize the convergent point for the general case using Jordan normal form of the matrix. Let $\boldsymbol{Z} = \boldsymbol{M}^\top\boldsymbol{D}_k(\boldsymbol{M} - \gamma\boldsymbol{N})$ and $\boldsymbol{Z} = \boldsymbol{Q}\boldsymbol{J}\boldsymbol{Q}^{-1}$ be the jordan normal form of $\boldsymbol{Z}$. We still denote $\boldsymbol{Q}_0$ the eigenvectors with eigenvalue zero. Then there is

$$\lim_{t\to\infty}(\boldsymbol{I} - \eta\boldsymbol{Z})^t = \lim_{t\to\infty}\boldsymbol{Q}(\boldsymbol{I} - \eta\boldsymbol{J})^t\boldsymbol{Q}^{-1} \tag{57}$$

Since $\boldsymbol{I} - \eta\boldsymbol{J}$ has a block diagonal structure, its power can be obtained by first computing the power of each block. Let $\boldsymbol{J}_i$ be the jordan block with eigenvalue $\lambda_i$. We write $\boldsymbol{J}_i = \lambda_i\boldsymbol{I} + \boldsymbol{L}$, where $\boldsymbol{L}$ is a matrix such that the only non-zero entries of $\boldsymbol{L}$ are on the first off-diagonal. Then we can write the $i$-th block of $\boldsymbol{J}$ as $(1 - \eta\lambda_i)\boldsymbol{I} - \eta\boldsymbol{L}$. Using the binomial theorem we get

$$((1 - \eta\lambda_i)\boldsymbol{I} - \eta\boldsymbol{L})^t = \sum_{s=0}^{t}\binom{t}{s}(1 - \eta\lambda_i)^{t-s}(-\eta\boldsymbol{L})^s. \tag{58}$$

Note that $\boldsymbol{L}^s$ is the matrix with ones on the $s$-th diagonal away from the main diagonal, and $\boldsymbol{L}^s = 0$ for $s$ larger than the size of $\boldsymbol{L}$. Therefore, $((1-\eta\lambda_i)\boldsymbol{I}-\eta\boldsymbol{L})^t$ is a triangular matrix with $(1-\eta\lambda_i)^t$ on the main diagonal, $-\eta t(1-\eta\lambda_i)^{t-1}$ on the first off-diagonal, and so on. Therefore, the eigenvalues of this matrix are all $(1-\eta\lambda_i)^t$. Then given a learning rate that $\eta < 1/\lambda_{\max}$, for any jordan block with $\lambda_i > 0$, we have that the matrix power converges. For $\lambda_i = 0$, the jordan block corresponds to eigenvectors that are in the kernel space of $\boldsymbol{Z}$. Thus, suppose that all eigenvalues of $\boldsymbol{Z}$ are non-negative, we have

$$\lim_{t\to\infty}\boldsymbol{Q}(\boldsymbol{I} - \eta\boldsymbol{J})^t\boldsymbol{Q}^{-1}\theta_0 = \boldsymbol{Q}_0\boldsymbol{Q}_0^{-1}\theta_0. \tag{59}$$

Note that if a negative $\lambda_i$ exists, the above derivations can still be used to characterize the convergent sub-component of $\theta_0$. The non-convergent sub-component of $\theta_0$ will diverge with an exponential rate as shown above.

$\square$

### A.3 Proof of Theorem 3

*Proof.* We first prove the update formula. Recall the FVI update,

$$\theta_t = \boldsymbol{M}^\dagger(\boldsymbol{R} + \gamma\boldsymbol{N}\theta_{t-1}).$$

For $t = 1$ the result holds by definition. Suppose that

$$\theta_t = \boldsymbol{M}^\dagger \left( \sum_{i=0}^{t-2} (\gamma \boldsymbol{N}\boldsymbol{M})^i \boldsymbol{R} + (\gamma \boldsymbol{N}\boldsymbol{M}^\dagger)^{t-1}(\boldsymbol{R} + \gamma \boldsymbol{N}\theta_0) \right) \tag{60}$$

We now prove the result for $t + 1$ by induction.

$$\theta_{t+1} = \boldsymbol{M}^\dagger(\boldsymbol{R} + \gamma \boldsymbol{N}\theta_t) \tag{61}$$

$$= \boldsymbol{M}^\dagger \left( \boldsymbol{R} + \gamma \boldsymbol{N}\boldsymbol{M}^\dagger \left( \sum_{i=0}^{t-2} (\gamma \boldsymbol{N}\boldsymbol{M}^\dagger)^i \boldsymbol{R} + (\gamma \boldsymbol{N}\boldsymbol{M}^\dagger)^{t-1}(\boldsymbol{R} + \gamma \boldsymbol{N}\theta_0) \right) \right) \tag{62}$$

$$= \boldsymbol{M}^\dagger \left( \boldsymbol{R} + \left( \sum_{i=1}^{t-1} (\gamma \boldsymbol{N}\boldsymbol{M}^\dagger)^i \boldsymbol{R} + (\gamma \boldsymbol{N}\boldsymbol{M}^\dagger)^t(\boldsymbol{R} + \gamma \boldsymbol{N}\theta_0) \right) \right) \tag{63}$$

$$= \boldsymbol{M}^\dagger \left( \sum_{i=0}^{t-1} (\gamma \boldsymbol{N}\boldsymbol{M}^\dagger)^i \boldsymbol{R} + (\gamma \boldsymbol{N}\boldsymbol{M}^\dagger)^t(\boldsymbol{R} + \gamma \boldsymbol{N}\theta_0) \right) \tag{64}$$

Clearly the convergence of this algorithm depends on the spectral norm of $\boldsymbol{N}\boldsymbol{M}^\dagger$. In particular, given that $\|\boldsymbol{N}\boldsymbol{M}^\dagger\| < 1/\gamma$, we have the algorithm converges to

$$\boldsymbol{M}^\dagger(\boldsymbol{I}_k - \gamma \boldsymbol{W})^{-1}\boldsymbol{R} \tag{65}$$

as $t \to \infty$. This finishes the proof. $\qquad\square$

## A.4 PROOF OF THEOREM 4 AND COROLLARY 1

*Proof.* We first prove the result for residual minimization fixed point $\theta_{\mathrm{RM}}$. The proof is adopted from characterizing the minimum norm solution of solving least square (Boyd and Vandenberghe, 2004). Let $\boldsymbol{A} = \boldsymbol{M} - \gamma \boldsymbol{N}$ for simplicity. We write the Lagrange of the optimization problem,

$$\mathcal{L}(\theta, \alpha) = \inf_{\theta \in \mathbb{R}^d} \sup_{\alpha \in \mathbb{R}^k} \frac{1}{2}\|\theta\|^2 + \alpha^\top(\boldsymbol{R} - \boldsymbol{A}\theta) \tag{66}$$

$$= \sup_{\alpha \in \mathbb{R}^k} \frac{1}{2}\|\boldsymbol{A}^\top\alpha\|^2 + \alpha^\top \boldsymbol{R} - \alpha^\top \boldsymbol{A}\boldsymbol{A}^\top\alpha \tag{67}$$

$$= \sup_{\alpha \in \mathbb{R}^k} \alpha^\top \boldsymbol{R} - \frac{1}{2}\alpha^\top \boldsymbol{A}\boldsymbol{A}^\top\alpha. \tag{68}$$

Solving for $\alpha^*$ and add it to $\theta^* = \boldsymbol{A}^\top\alpha^*$ gives that $\theta^* = \boldsymbol{A}^\dagger \boldsymbol{R}$.

We next prove Corollary 1, which characterizes the TD and FVI fixed point $\theta_{\mathrm{TD}}$. Let $\boldsymbol{W} = \boldsymbol{N}\boldsymbol{M}^\dagger$. We write the Lagrange of the optimization problem,

$$\mathcal{L}(\theta, \alpha) = \inf_{\theta \in \mathbb{R}^d} \sup_{\alpha \in \mathbb{R}^k} \frac{1}{2}\|\theta\|^2 + \alpha^\top(\boldsymbol{R} - (\boldsymbol{I}_k - \gamma \boldsymbol{W})\boldsymbol{M}\theta) \tag{69}$$

$$= \sup_{\alpha \in \mathbb{R}^k} \frac{1}{2}\|\boldsymbol{M}^\top(\boldsymbol{I}_k - \gamma \boldsymbol{W})^\top\alpha\|^2 + \alpha^\top \boldsymbol{R} - \alpha^\top(\boldsymbol{I}_k - \gamma \boldsymbol{W})\boldsymbol{M}\boldsymbol{M}^\top(\boldsymbol{I}_k - \gamma \boldsymbol{W})^\top\alpha \tag{70}$$

$$= \sup_{\alpha \in \mathbb{R}^k} \alpha^\top \boldsymbol{R} - \frac{1}{2}\alpha^\top(\boldsymbol{I}_k - \gamma \boldsymbol{W})\boldsymbol{M}\boldsymbol{M}^\top(\boldsymbol{I}_k - \gamma \boldsymbol{W})^\top\alpha. \tag{71}$$

Solving for $\alpha^*$ and add it to $\theta^* = \boldsymbol{M}^\top(\boldsymbol{I}_k - \gamma \boldsymbol{W})^\top\alpha^*$ gives that $\theta^* = \boldsymbol{M}^\dagger(\boldsymbol{I}_k - \gamma \boldsymbol{W})^{-1}\boldsymbol{R}$. The second part of Theorem 4 is immediately followed by this.

$$\square$$

## A.5 PROOF OF THEOREM 5

**Lemma 2.** *Let $\theta_t$ be the output of FVI at iteration $t$ with $\theta_0$ as the initial parameter. We have that $\boldsymbol{M}^\dagger \boldsymbol{M}\theta_t = \theta_t$ for any $t \geq 1$.*

*Proof.* The claim is implied by the fact that $\theta_t$ is in the row space of $\boldsymbol{M}$. In particular, by Theorem 3, $\theta_t = \boldsymbol{M}^\dagger \alpha$ for some $\alpha \in \mathbb{R}^n$. Thus,

$$\boldsymbol{M}^\dagger \boldsymbol{M} \theta_t = \boldsymbol{M}^\dagger \boldsymbol{M} \boldsymbol{M}^\dagger \alpha = \boldsymbol{M}^\dagger \alpha = \theta_t \,. \tag{72}$$

This finishes the proof. □

**Lemma 3.** $\mathcal{E}(\theta)$ *is 1-smoothness.*

*Proof.* Recall the prediction error of $\theta \in \mathbb{R}^d$

$$\mathcal{E}(\theta) = \frac{1}{2} \|\Phi\theta - v\|_{\boldsymbol{D}_\mu}^2 = \frac{1}{2} \|\theta - \theta^*\|_\Sigma^2 \,, \tag{73}$$

where $\Sigma = \Phi^\top \boldsymbol{D}_\mu \Phi$. The gradient of $\theta$ is $\mathcal{E}'(\theta) = \Sigma(\theta - \theta^*)$. Then

$$\|\mathcal{E}'(\theta_1) - \mathcal{E}'(\theta_2)\| = \|\Sigma(\theta_1 - \theta_2)\| \le \lambda_{\max}(\Sigma) \|(\theta_1 - \theta_2)\| \le \|\theta_1 - \theta_2\| \,, \tag{74}$$

where we use $\|\phi(s)\| \le 1$ for all $s \in \mathcal{S}$ and $\lambda_{\max}(\Sigma) \le \sum_s \mu(s) \lambda_{\max}(\phi(s)\phi(s)^\top) \le 1$. □

**Lemma 4.** *Let* $\varepsilon_{app} = \boldsymbol{N}\Pi_{\boldsymbol{M}}^\perp \theta^*$ *and* $\sigma_{stat} = \boldsymbol{H}(P - \hat{P})\Phi\theta^*$. *We have*

$$\boldsymbol{M}\theta^* = \boldsymbol{R} + \gamma(\varepsilon_{app} + \varepsilon_{stat}) + \gamma\boldsymbol{W}\boldsymbol{M}\theta^* \,. \tag{75}$$

*Proof.* Using the definitions we have,

$$\boldsymbol{M}\theta^* = \boldsymbol{R} + \gamma\boldsymbol{H}P\Phi\theta^* \tag{76}$$

$$= \boldsymbol{R} + \gamma\boldsymbol{N}\theta^* + \gamma\boldsymbol{H}(P - \hat{P})\Phi\theta^* \tag{77}$$

$$= \boldsymbol{R} + \gamma(\boldsymbol{W}\boldsymbol{M} + \boldsymbol{N}\Pi_{\boldsymbol{M}}^\perp)\theta^* + \gamma\boldsymbol{H}(P - \hat{P})\Phi\theta^* \tag{78}$$

$$= \boldsymbol{R} + \gamma(\varepsilon_{\text{app}} + \varepsilon_{\text{stat}}) + \gamma\boldsymbol{W}\boldsymbol{M}\theta^* \,. \tag{79}$$

□

*Proof.* By Theorem 3, $\theta_t = \boldsymbol{M}^\dagger \mathcal{T}^{t-1}(\boldsymbol{R} + \gamma\boldsymbol{N}\theta_0)$ is the output of FVI at iteration $t$. By noting that $\mathcal{E}(\theta^*) = 0$ and Lemma 3, for any $\theta \in \mathbb{R}^d$.

$$\mathcal{E}(\theta) \le \frac{1}{2}\|\theta - \theta^*\|^2 = \frac{1}{2}\left(\|\theta - \theta^*\|_{\boldsymbol{M}^\dagger \boldsymbol{M}}^2 + \|\theta - \theta^*\|_{\boldsymbol{I}_d - \boldsymbol{M}^\dagger \boldsymbol{M}}^2\right) \,. \tag{80}$$

We first consider the second term. By Lemma 2,

$$\|\theta_t - \theta^*\|_{\boldsymbol{I}_d - \boldsymbol{M}^\dagger \boldsymbol{M}}^2 = (\theta_t - \theta^*)^\top \left(\boldsymbol{I}_d - \boldsymbol{M}^\dagger \boldsymbol{M}\right)(\theta_t - \theta^*) = \|\theta^*\|_{\boldsymbol{I}_d - \boldsymbol{M}^\dagger \boldsymbol{M}}^2 \,. \tag{81}$$

We now consider the first term. By Lemma 4,

$$\boldsymbol{M}(\theta^* - \theta_t) = \boldsymbol{M}\theta^* - \mathcal{T}^{t-1}(\boldsymbol{R} + \gamma\boldsymbol{N}\theta_0) \tag{82}$$

$$= \sum_{i=0}^{t-2}(\gamma\boldsymbol{W})^i\left(\boldsymbol{R} + \gamma(\varepsilon_{\text{app}} + \varepsilon_{\text{stat}})\right) + (\gamma\boldsymbol{W})^{t-1}(\boldsymbol{M}\theta^*) - \sum_{i=0}^{t-2}(\gamma\boldsymbol{W})^i\boldsymbol{R} - (\gamma\boldsymbol{W})^{t-1}(\boldsymbol{R} + \gamma\boldsymbol{N}\theta_0) \tag{83}$$

$$= \gamma\left(\sum_{i=0}^{t-2}(\gamma\boldsymbol{W})^i\varepsilon_{\text{app}} + \sum_{i=0}^{t-1}(\gamma\boldsymbol{W})^i\varepsilon_{\text{stat}} + (\gamma\boldsymbol{W})^{t-1}\boldsymbol{N}(\theta^* - \theta)\right) \tag{84}$$

Let $\hat{\Sigma} = \boldsymbol{M}^\top \boldsymbol{D}_k \boldsymbol{M}$ be the empirical covariance matrix. Note that $\lambda_{\min}(\boldsymbol{M}^\top \boldsymbol{M})/k \ge \lambda_{\min}(\hat{\Sigma})$. To show this, let $\bar{\boldsymbol{D}} = \text{diag}(\frac{1}{k}, \ldots, \frac{1}{k})$, and $\boldsymbol{M} = \boldsymbol{U}\boldsymbol{S}\boldsymbol{V}^\top$ be the SVD of $\boldsymbol{M}$. Then

$$\lambda_{\min}^+(\boldsymbol{M}^\top \bar{\boldsymbol{D}}\boldsymbol{M}) = \min_{\|x\|=1} x^\top \boldsymbol{M}^\top \bar{\boldsymbol{D}}\boldsymbol{M}x \tag{85}$$

$$= \min_{\|\alpha\|=1} \alpha^\top \boldsymbol{S}\boldsymbol{U}^\top \bar{\boldsymbol{D}}\boldsymbol{U}\boldsymbol{S}\alpha \tag{86}$$

$$\ge \min_{\|\alpha\|=1} \alpha^\top \boldsymbol{S}\boldsymbol{U}^\top \boldsymbol{D}_k \boldsymbol{U}\boldsymbol{S}\alpha \tag{87}$$

$$= \lambda_{\min}^+(\boldsymbol{M}^\top \boldsymbol{D}_k \boldsymbol{M}) \tag{88}$$

where we replace $x = V\alpha$ since $V$ are orthonormal bases, and $\min_{i \in [k]} \hat{\mu}_i \leq 1/k$. Therefore,

$$\|M^\dagger\| = 1/\sqrt{\lambda^+_{\min}(M^\dagger M)} \leq 1/\sqrt{k\lambda^+_{\min}(\hat{\Sigma})}.$$

Combining the results above we have,

$$\|\theta_t - \theta^*\|^2_{M^\dagger M} = \left\|M^\dagger M(\theta_t - \theta^*)\right\|^2 \tag{89}$$

$$\leq \|M^\dagger\|^2 \|M(\theta_t - \theta^*)\|^2 \tag{90}$$

$$\leq \frac{\gamma}{k\lambda_{\min}(\hat{\Sigma})} \left\|\sum_{i=0}^{t-1}(\gamma W)^i(\varepsilon_{\text{app}} + \varepsilon_{\text{stat}}) + (\gamma W)^{t-1}N(\theta^* - \theta)\right\|^2 \tag{91}$$

$$\leq \frac{4\gamma}{k\lambda_{\min}(\hat{\Sigma})} \left((\varepsilon^2 + \sigma^2)\left\|\sum_{i=0}^{t-1}(\gamma W)^i\right\|^2 + \left\|(\gamma W)^{t-1}\right\|^2 \|\Phi\|^2\|\theta_0 - \theta^*\|^2\right). \tag{92}$$

Combine this with Eq. (81) finishes the proof. $\qquad \square$

## A.6 Proof of Corollary 2

*Proof.* Recall that in the proof of Theorem 5 we have

$$\|\theta_t - \theta^*\|^2_{M^\dagger M} \leq \frac{4\gamma}{k\lambda_{\min}(\hat{\Sigma})} \left((\varepsilon^2 + \sigma^2)\left\|\sum_{i=0}^{t-1}(\gamma W)^i\right\|^2 + \left\|(\gamma W)^{t-1}\right\|^2 \|\Phi\|^2\|\theta_0 - \theta^*\|^2\right). \tag{93}$$

Given that $\|W\| < 1$, we have for the fixed point $\theta_\infty$,

$$\|\theta_t - \theta^*\|^2_{M^\dagger M} \leq \frac{4\gamma(\varepsilon^2 + \sigma^2)}{k\lambda_{\min}(\hat{\Sigma})(1 - \gamma)^2} \tag{94}$$

We first consider $\sigma^2 = \|H(P - \hat{P})v\|^2$. By Hoeffding's inequality and a union bound we have with probability at least $1 - \delta$, for any $s \in \text{supp}(D)$,

$$\left|(\hat{P}_s - P_s)^\top v\right| \leq \frac{1}{1 - \gamma}\sqrt{\frac{\log(|\mathcal{S}|/\delta)}{2n(s)}}. \tag{95}$$

Thus, let $n_{\min} = \min_{s:n(s)>0} n(s)$, we have

$$\frac{\sigma^2}{k} \leq \frac{\log(|\mathcal{S}|/\delta)}{2(1 - \gamma)^2 n_{\min}}. \tag{96}$$

Now we consider $\varepsilon^2 = \|N\Pi^\perp_M \theta^*\|^2$. Since $N\Pi^\perp_M$ is perpendicular to $M$, and all features have norm bounded by one,

$$\frac{\varepsilon^2}{k} \leq \|\theta^*\|^2_{I_d - M^\dagger M}. \tag{97}$$

Combine the above we have,

$$\mathcal{E}(\theta) \leq \frac{1}{2}\|\theta - \theta^*\|^2_{M^\dagger M} + \frac{1}{2}\|\theta^*\|^2_{I_d - M^\dagger M} \tag{98}$$

$$\leq \frac{2\gamma}{\lambda_{\min}(\hat{\Sigma})(1 - \gamma)^2}\left(\frac{\log(|\mathcal{S}|/\delta)}{2(1 - \gamma)^2 n_{\min}} + \|\theta^*\|^2_{I_d - M^\dagger M}\right) + \frac{1}{2}\|\theta^*\|^2_{I_d - M^\dagger M} \tag{99}$$

$$= \frac{\gamma\log(|\mathcal{S}|/\delta)}{\lambda_{\min}(\hat{\Sigma})(1 - \gamma)^4 n_{\min}} + \frac{4\gamma}{\lambda_{\min}(\hat{\Sigma})(1 - \gamma)^2}\|\theta^*\|^2_{I_d - M^\dagger M} \tag{100}$$

Finally, using the tower rule gives the desired result.

$\square$

## A.7 Concentration of Eigenvalues and Bounding the Orthogonal Complement

We will need the following result (Kuzborskij et al., 2021, Theorem 6), which is concerned with the magnitude of projection onto the eigenspace of a covariance matrix. The result is based on (Shawe-Taylor et al., 2005, Theorem 1)

**Lemma 5.** *Let* $\hat{\Sigma} = \frac{1}{n} \sum_i x_i x_i^\top$ *be the covariance matrix of i.i.d. data* $x_i \in \mathbb{R}^d$*. Denote the* $h$*-"tail" of eigenvalues of a covariance matrix* $\hat{\Sigma} =$ *as*

$$\lambda^{>h} = \sum_{i=h+1}^{n} \lambda_i \,. \tag{101}$$

*Let* $\boldsymbol{U}_r$ *be the first* $r$ *eigenbasis for* $r \in [n]$*. Then for any* $z \in \mathbb{R}^d$*, with probability at least* $1 - \delta$*,*

$$\mathbb{E}\left[\|\Pi_{\boldsymbol{U}_r}^\perp z\|_2^2\right] \leq \min_{h \in [r]} \left\{ \frac{1}{n}\lambda^{>h} + \frac{1+\sqrt{h}}{\sqrt{n}}\sqrt{\frac{2}{n}\sum_{i=1}^{n}\|x_i\|^2} \right\} + \|z\|_2^2 \sqrt{\frac{18}{n}\ln\left(\frac{2n}{\delta}\right)}\,. \tag{102}$$

The next lemma gives a non-asymptotic result to understand the behaviour of $\hat{\lambda}_{\min}$ (Kuzborskij et al., 2021, Lemma 1).

**Lemma 6.** *Let* $\boldsymbol{X} = [\boldsymbol{X}_1, \ldots, \boldsymbol{X}_n] \in \mathbb{R}^{d \times n}$ *be a random matrix with i.i.d. columns, such that* $\max_i \|\boldsymbol{X}_i\|_2 \leq K$*, and let* $\hat{\Sigma} = \boldsymbol{X}\boldsymbol{X}^\top/n$*, and* $\Sigma = \mathbb{E}[\boldsymbol{X}_1\boldsymbol{X}_1^\top]$*. Then, for every* $\alpha \geq 0$*, with probability at least* $1 - 2e^{-\alpha}$*, we have*

$$\lambda_{\min}^+(\hat{\Sigma}) \geq \lambda_{\min}^+(\Sigma)\left(1 - K^2\left(c\sqrt{\frac{d}{n}} + \sqrt{\frac{\alpha}{n}}\right)\right)_+^2 \quad \text{for } n \geq d\,, \tag{103}$$

*and furthermore, assuming that* $\|\boldsymbol{X}_i\|_{\Sigma^\dagger} = \sqrt{d}$*, for all* $i \in [n]$*, we have*

$$\lambda_{\min}^+(\hat{\Sigma}) \geq \lambda_{\min}^+(\Sigma)\left(\sqrt{\frac{d}{n}} - K^2\left(c + 6\sqrt{\frac{\alpha}{n}}\right)\right)_+^2 \quad \text{for } n < d\,, \tag{104}$$

*where we have an absolute constant* $c = 2^{3.5}\sqrt{\ln 9}$*.*

## B Experiment setup

In this section, we provide additional details about the experimental setup and hyper-parameters used for each of the environments. For all of these environments the regularization weights were considered as tunable hyper-parameters. In addition, for $R_\phi$ (see Eq 23), the scale factor $\beta$ was also considered as a parameter to be tuned in order to approximate the feature matrix norm.

### B.1 Acrobot

- Replay buffer with 10k tuples sampled using a random policy across trajectories with maximum episode length of 64.
- A DQN with hidden units consisting of fully connected layers with $(100, 100)$ units.
- Batch size 64.
- Learning rate of 1e-3.
- Regularized RM with weight of 2e-2 on $\mathcal{R}_\phi$ and 1e-4 on $\mathcal{R}_w$.
- Regularized TD with weight of 0 on $\mathcal{R}_\phi$ and 1e-4 on $\mathcal{R}_w$.

### B.2 Reacher

- Replay buffer with 10k tuples sampled from a random policy across trajectories with maximum steps per episode of length 50.

- Learning rate 1e-4.
- A value network for the continuous action inputs with a fc observation layer with params (50,), a fc action layer with params (50,) and a joint fc layer with params (100,).
- Batch size 64.
- Gradient clipping with a norm of 10.0
- Regularized RM with weight of 0.15 on $\mathcal{R}_w$ and 0 on $\mathcal{R}_\phi$.
- Regularized TD with weight of 2e-2 on $\mathcal{R}_w$ and 7e-3 on $\mathcal{R}_\phi$.

### B.3 CARTPOLE

- Replay buffer with 10k tuples sampled using a random policy across trajectories with maximum steps per episode of length 50.
- A DQN with hidden units consisting of fully connected layers with $(100, 100)$ units.
- Batch size 64.
- Learning rate 1e-3.
- Regularized RM with weight of 0.29 on $\mathcal{R}_w$ and 0 on $\mathcal{R}_\phi$.
- Regularized TD with weight of 1.5e-3 on $\mathcal{R}_w$ and 5e-3 on $\mathcal{R}_\phi$.

### B.4 PENDULUM

- Replay buffer with 1k tuples obtained by sampling directly from a fixed initial state distribution using a random policy.
- A value network for the continuous action inputs with a fc observation layer with params (50,), a fc action layer with params (50,) and a joint fc layer with params (100,).
- Batch size 64.
- Learning rate 1e-3.
- Regularized RM with weight of 1.0 on $\mathcal{R}_w$ and 5.4e-4 on $\mathcal{R}_\phi$.
- Regularized TD with weight of 0 on $\mathcal{R}_w$ and 1.0 on $\mathcal{R}_\phi$.

### B.5 EXTRA EXPERIMENT RESULTS

We provide extra experiment results on four Mujoco control problems to assess the applicability of the proposed regularization $R_\phi$: HalfCheetah, Hopper, Ant, and Walker2d. The results are provided in Fig. 3. All results are averaged over 100 runs with different random seeds. The hyper-parameters are provided below.

- The Q-function is approximated by two hidden layer fully neural networks, where the hidden layer size is 256.
- Batch size 256.
- Learning rate 3e-4.
- Regularized TD weight are tuned from $\{1e-4, 1e-3, 1e-2, 1e-1, 1\}$

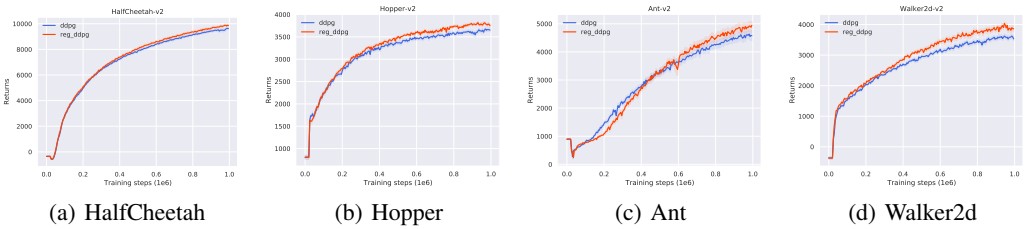

Figure 3: *We show the results with proposed regularization compared to the baseline algorithms. The algorithms are trained using a fixed offline data set collected by random initialized policies. The x-axis shows the training iterations (in thousands) and y-axis shows the performance. All plots are averaged over 100 runs. The shaded area shows the standard error.*

