# OpenReview forum: "Understanding and Leveraging Overparameterization in Recursive Value Estimation"
_ICLR.cc/2022/Conference — ICLR 2022 Poster_

### Official Review · Reviewer_gT5o · 2021-10-26

**Correctness:** 3
**Technical Novelty And Significance:** 2
**Empirical Novelty And Significance:** 2
**Recommendation:** 5
**Confidence:** 3

**Main Review:**

1. This paper is concerned with the overparameterized linear representations of TD, FVI, and RM. The paper is easy to follow and digest. The overparameterized representation problem is a crucial problem. On the downside, the linear case is restricted, and technical analysis is not very novel. I have some high-level comments as follows.

2. The paper discusses the unified view of overparameterized linear representation. Does such view also hold for the underparameterized linear representation? If not, is there any intuition that why we would have such difference?

3. For the overparameterized problem, I think non-linear representation like deep neural network is more common and interesting. Although the linear representation result in this paper is new to me, it is too restricted. Usually we would not use linear function approximation with lots of parameters, therefore I feel the result in this paper has limited impact. Is it possible to extend to neural network with Relu activation?

4. Related to the prior point, I suggest the authors to add some discussions about the related work of overparameterized NN theory, e.g., NTK.

5. This paper defines the update equation for the underparameterized form (5,6) and says the equation for the overparameterized case (10,12) can be directly obtained. Can you provide some details of the derivation?

6. Is there any relationship between the solution of TD, FVI, and RM with overparameterized linear representation and the overparameterized linear regression?

7. What is the intuition to define D matrix in proposition 1?

8. Is there any interpretation of operator M, N, and R in the overparameterized regime? It seems that they can be obtained by pre-multiplying H matrix from the original operator for the underparameterized case? If so, why pre-multiplying H would be useful here?

9. For the overparameterized regime (theorem 4), it seems that the difference between TD/FVI and RM is that the convergence points of TD/FVI further constraints that \theta must lie in the row span of M. This is quite similar as the interpretation that TD/FVI is the solution of MSPBE while RM is the solution of MSBE (see ref) because the objective of MSPBE and MSBE are quite similar and they only differ by a projection operator. This projection operator is similar as the forcing \theta to lie in the row span of M. Is my understanding correct?

Ref: Policy Evaluation with Temporal Differences: A Survey and Comparison

10. In the theorem statement, it is mentioned that the algorithm will not converge if the norm is large. Could you provide more discussions about the implication?

11. The font size in the figure is too small.

**Summary Of The Paper:**

In this paper, the authors consider the overparameterized linear representations of TD, FVI, and RM. A unified interpretation of these algorithms of minimizing the Euclidean norm of the weights subject to alternative constraints is proposed. The paper is also supported by the empirical results.

**Summary Of The Review:**

This is a theoretical paper about the overparameterized linear representations of TD, FVI, and RM. This is an interesting question. On the downside, the linear representation might be too restricted and the technical contribution is not significant.

---

> ### Author Response · Authors · 2021-11-16
> **Discussion with Reviewer gT5o (1/2)**
>
> We thank the reviewer for the work and input. The detailed responses are provided below.
>
> **The linear representation result is too restricted. Is it possible to extend to neural network with Relu activation? NTK.**
>
> The connection between an overparameterized linear model and deep neural network can be established using neural tangent kernels. In Appendix A.8 of the revision, we show how to derive the convergence of FVI using the neural tangent kernel as an example. Here we briefly discuss the idea.
>
> Let $v(s, \omega)$ be the value function approximated by a neural network, where $\omega$ is all the parameters of the neural network.
> The neural tangent kernel is defined as
> \begin{align}
>     \text{ker}(s, s') =
>     \mathbb{E}_{\omega\sim\Omega}
>     \left\langle
>     \frac{\partial v(s, \omega)}{\partial \omega},
>     \frac{\partial v(s', \omega)}{\partial \omega}
>     \right\rangle
> \end{align}
> where $s, s'$ are two states, $\Omega$ is an initialization distribution over $\omega$.
> Let $\omega_0$ be random initial parameters sampled from $\Omega$, there is
> \begin{align}
>     \text{ker}_0(s,s') = \left\langle
>     \frac{\partial v(s, \omega_0)}{\partial \omega_0},
>     \frac{\partial v(s', \omega_0)}{\partial \omega_0}
>     \right\rangle
>     \approx
>     \text{ker}(s, s')\, .
> \end{align}
> This approximation is accurate when the width of the neural network is allowed to increase to infinity.
> See Theorem 3.1 in [2] for the proof. Note that in this kernel, an input $s$ is mapped to a feature vector $\phi(s) = \frac{\partial v(s, \omega_0)}{\partial \omega_0}$, which is a $d$-dimensional linear feature where $d$ is the total number of parameters of the neural network. The main results of this paper can be extended to this setting using the kernel trick.
>
>
> Finally, since the overparameterized linear setting is new for the reviewer, we would like to point out that there are some recent works that study supervised learning with the “simple” linear models and provide very insightful results for better understanding the overparameterization regime [1,2,3].
>
>
> **Does such view also hold for the underparameterized linear representation? If not, is there any intuition that why we would have such difference?**
>
> We are not aware of a similar unified view in the underparameterized setting. The reason is that in that setting one cannot exactly drive the Bellman error to zero due to the approximation error.
>
>
> **Eq (10) and (12)**
>
> We show how to obtain Equation 12 from 6. Equation 10 can be derived similarly.  Basically one just needs to use Proposition 1 and plug-in the definitions of M, N and R.  In particular,  the TD update is
> \begin{align*}
>     \theta_{t+1} = \theta_t - \eta \Phi^\top D (\Phi \theta_t - (r + \gamma \hat{P} \Phi \theta_t))\.
> \end{align*}
>
> Plug in $D = H^\top D_k H$,
> \begin{align*}
>     \theta_{t+1}  = \theta_t - \eta \Phi^\top H^\top D_k H (\Phi \theta_t - (r + \gamma \hat{P} \Phi \theta_t))
>     = \theta_t - \eta M^\top D_k (M - \gamma N) \theta_t
>     + \eta M^\top D R \.
> \end{align*}
>
>
> **Define D matrix in proposition 1**
>
> The D matrix is not defined in Proposition 1. It is the empirical data distribution defined in Section 3.1. The usage of the data distribution matrix is standard in RL theory, see [4,5,6,7] and Chapter 9-11 in [8] for example. With this matrix, one can apply compact matrix operation in the derivations to simplify the derivations.
>
>
> **Is there any interpretation of operator M, N, and R...**
>
> We do not understand this question. What does it mean “they can be obtained by pre-multiplying H matrix from the original operator for the underparameterized case”?
>
> The definitions of M, N and R are clearly made in Section 4, where we can compactly represent the batch dataset with these three matrices.
>
> The intuition behind introducing $H$ and $D_k$ is to simplify the derivations of the main results. For example, instead of using M, one can define a $n\times d$ feature matrix, where the $i$-th row corresponds to the feature of $s_i$ in the data set. However, in the data set we might have duplicated states, which makes such a feature matrix low rank. This might make the derivations much more complicated as one need to work on the low rank matrix at each step. We explicitly avoid this by introducing $H$ and $D_k$.
>
> **This is quite similar as the interpretation...**
>
>
> The reviewer seems to suggest that the characterization of the fixed points can be directly implied by standard results in the underparameterized setting. This is definitely incorrect, as the
> MSPBE and MSBE solutions actually collapse when there is no approximation error. Thus, the classical view in the underparameterized setting fails to explain the algorithmic bias in the overparameterized setting. We emphasize that it is indeed the contribution of this very paper to explicitly characterize the fixed points of different algorithms, from which the unified view can be established.

---

> ### Author Response · Authors · 2021-11-16
> **References**
>
> [1] Bartlett, Peter L., Andrea Montanari, and Alexander Rakhlin. "Deep learning: a statistical viewpoint." arXiv preprint arXiv:2103.09177 (2021).
>
> [2] Bartlett, Peter L., Philip M. Long, Gábor Lugosi, and Alexander Tsigler. "Benign overfitting in linear regression." Proceedings of the National Academy of Sciences 117, no. 48 (2020): 30063-30070.
>
> [3] Kuzborskij, Ilja, Csaba Szepesvári, Omar Rivasplata, Amal Rannen-Triki, and Razvan Pascanu. "On the role of optimization in double descent: A least squares study." arXiv preprint arXiv:2107.12685 (2021).
>
> [4] Tsitsiklis, John N., and Benjamin Van Roy. "An analysis of temporal-difference learning with function approximation." IEEE transactions on automatic control 42, no. 5 (1997): 674-690.
>
> [5] Kolter, J. "The fixed points of off-policy TD." Advances in Neural Information Processing Systems 24 (2011): 2169-2177.
>
> [6] Dann, Christoph, Gerhard Neumann, and Jan Peters. "Policy evaluation with temporal differences: A survey and comparison." Journal of Machine Learning Research 15 (2014): 809-883.
>
> [7] Bhandari, Jalaj, Daniel Russo, and Raghav Singal. "A finite time analysis of temporal difference learning with linear function approximation." In Conference on learning theory, pp. 1691-1692. PMLR, 2018.
>
> [8] Sutton, Richard S., and Andrew G. Barto. Reinforcement learning: An introduction. MIT press, 2018.

---

> ### Author Response · Authors · 2021-11-24
> **Discussion**
>
> We are wondering if our responses addressed your concerns. We will be happy to answer if there are additional issues/questions.

---

> > ### Comment · Reviewer_gT5o · 2021-11-25
> > **Response**
> >
> > Thank you for the response, and it helps me understand the paper better.
> >
> > 7. The question was not clear. I was asking the intuition to decompose D matrix in proposition 1.
> >
> > 8. This might not be a good question. Yes I understand they are clearly defined. I'm just curious why the difference between over and under para case seem to be an additional H factor (quote: It seems that they can be obtained by pre-multiplying H matrix from the original operator for the underparameterized case).

---

> > > ### Author Response · Authors · 2021-11-25
> > > **Response**
> > >
> > > Thanks for the response. We hope the reviewer's question can be addressed below. We would like to discuss this further if the reviewer still has questions about this issue.
> > >
> > > **"I'm just curious why the difference between over and under para case seem to be an additional H factor"**
> > >
> > > We think the reviewer is referring to the difference between (5) and (11), (6) and (12). Please correct us if this is wrong.
> > >
> > > The algorithm update formulas (5) and (6) do not change from under to over parameterization. The difference occurs when studying the convergence of the updates. For example, consider the TD update (6). In the underparameterized regime, $\Phi^\top D \Phi$ is invertible. Thus one can derive the fixed point as shown in (7). By contrast, in the overparameterized regime, the matrix is not invertible as $d>|supp(\hat{\mu})|$. Then the research problem is how to prove the convergence of the algorithm under the low rank structure.
> > >
> > > The derivations in Theorem 1 and 2 exactly solve this problem. $H$ and $D_k$ are introduced to make the derivation under this low rank structure more easily. The mask matrix $H$ gives the unique states in the dataset. The $D_k$ matrix characterizes the empirical data distribution over these unique states. Thus, one does not need to handle the difficulty caused by the duplicated states/transitions in the derivation. Please also see our previous response for this point.

---

> > > > ### Author Response · Authors · 2021-12-02
> > > > **Response**
> > > >
> > > > We are wondering if your question has been answered yet. Please let me know if you need more details.

---

> > > > > ### Comment · Reviewer_gT5o · 2021-12-05
> > > > > **Thanks**
> > > > >
> > > > > Thank you for the response, and it helps me understand the paper better.

---

### Official Review · Reviewer_xq3y · 2021-10-27

**Correctness:** 2
**Technical Novelty And Significance:** 2
**Empirical Novelty And Significance:** 2
**Recommendation:** 3
**Confidence:** 4

**Main Review:**

Summary:
In this paper, the authors investigate the convergence results of three algorithms (RM, TD, and FVI) in OPE when the model is assumed to be over-parameterized linear function class. Besides, some empirical experiments are also conducted. But as I’m not a fully expert in empirical RL, I would not have any comments on the experiment part. To my opinion, the contribution of this work is not solid enough, even some results might be wrong. Please check the following parts.

Questions:
1.	This paper is claimed to understand over-parameterized model in OPE. However, after I read this paper, I’m still confused as there is no theoretical comparison between over- and under-parameterization cases. Besides, I also can’t learn from Thm 5 whether the size of model (d in this paper) has an effect on the learning process. Thus, I would be much appreciated if the authors could have a clear explanation of the motivation.

Errors:
1.	Equation (38), as t already goes infinity, RHS should be free of t.
2.	Equation (42), it should be $\theta_t$ at RHS. Thus, the recursive expression (43) is also wrong. (no effect on results I guess)
3.	At the bottom of page 15, it mentions that $M^\top D_k(M-\gamma N)$ can be eigen decomposed. Please explain why? To my knowledge, it’s not symmetric. And I think this would definitely have an impact on your results.
4.	In the proof of Corollary 2, Hoeffding’s inequality is applied to bound $\varepsilon$ and $\sigma$, and then taking an expectation over both sides in (97) without considering the high probability term $1-\delta$ when applying the Hoeffding’s inequality. I think this is definitely wrong.

Recommendation:
1.	To my opinion, the term ‘over-parameterization’ is confusing in this paper. To my best acknowledgement, the basic structure of an over-parameterized model is assumed a two-layer model, though a linear model is ok but not just enough.
2.	Appendix A.7 presents two lemmas to control the expectation term in Corollary 2. Please give readers exact form in your case not just present its original form.
3.	I think an improvement on writing is necessary. There are a number of grammatical problems and the language is not precise enough. Please have proofread.


**Summary Of The Paper:**

See main review.

**Summary Of The Review:**

See main review.

---

> ### Author Response · Authors · 2021-11-16
> **Discussion with Reviewer xq3y (1/2)**
>
> Thanks for carefully checking the derivations and pointing out typos. However, we believe that our derivation is correct, and the reviewer is likely overlooking our contributions.
>
> **Dependence on t in Equation 38 and 42**
>
> These are minor typos and we have fixed them in the revision, thanks.
>
> **Matrix eigendecomposition**
>
> As long as the matrix is a diagonalizable square matrix, it is eigendecomposable (https://en.wikipedia.org/wiki/Eigendecomposition_of_a_matrix). The symmetry is not required. We emphasize that this has no effect on the result of Theorem 2, where we consider $\theta_0 = 0$ and the derivation at the bottom of page 15 is not needed. All of the main contributions are unaffected by this observation. The diagonalizable assumption is added in the revision.
>
>
> **Error in the proof of Corollary 2**
>
> Thanks for spotting this. In Corollary 2, there is a missing term due to the small probability event. For example, if one assumes that the value function is bounded by $[0, V_\max]$, we have an extra term $\delta V_{\max}$ according to the small probability event in Eq 21 by the tower rule. This minor error does not affect the main result, and has been fixed in the revision. Also, we do not use Hoeffding’s inequality to bound the $\varepsilon$ term, which is actually bounded according to the geometric property (see Eq 94).
>
>
> **To my opinion, the term ‘over-parameterization’ is confusing in this paper. To my best acknowledgement, the basic structure of an over-parameterized model is assumed a two-layer model**
>
> The reviewer seems to have some fundamental misunderstanding of the definitions. Overparameterization refers to the regime where one can exactly fit the training data. In the value estimation problem considered by this paper, this means that one can exactly satisfy the temporal differences on all transitions in the batch data set, achieving zero Bellman error. This definition is clearly made in the paper (Section 3.1).
>
> The connection between an overparameterized linear model and deep neural network can be established using neural tangent kernels. In Appendix A.8 of the revision, we show how to derive the convergence of FVI using the neural tangent kernel as an example. Here we briefly discuss the idea.
>
> Let $v(s, \omega)$ be the value function approximated by a neural network, where $\omega$ is all the parameters of the neural network.
> The neural tangent kernel is defined as
> \begin{align}
>     \text{ker}(s, s') =
>     \mathbb{E}_{\omega\sim\Omega}
>     \left\langle
>     \frac{\partial v(s, \omega)}{\partial \omega},
>     \frac{\partial v(s', \omega)}{\partial \omega}
>     \right\rangle
> \end{align}
> where $s, s'$ are two states, $\Omega$ is an initialization distribution over $\omega$.
> Let $\omega_0$ be random initial parameters sampled from $\Omega$, there is
> \begin{align}
>     \text{ker}_0(s,s') = \left\langle
>     \frac{\partial v(s, \omega_0)}{\partial \omega_0},
>     \frac{\partial v(s', \omega_0)}{\partial \omega_0}
>     \right\rangle
>     \approx
>     \text{ker}(s, s')\, .
> \end{align}
> This approximation is accurate when the width of the neural network is allowed to increase to infinity.
> See Theorem 3.1 in [2] for the proof. Note that in this kernel, an input $s$ is mapped to a feature vector $\phi(s) = \frac{\partial v(s, \omega_0)}{\partial \omega_0}$, which is a $d$-dimensional linear feature where $d$ is the total number of parameters of the neural network. The main results of this paper can be extended to this setting using the kernel trick.
>
> **Though a linear model is ok but not just enough**
>
> The reviewer suggests that a linear model is not “enough”. In fact, there are some recent works that study supervised learning with the “simple” linear models and provide very insightful results for better understanding the overparameterization regime[1,2,3]. It will be very helpful if the reviewer can share with us your thoughts on these previous works and inform us what kind of result is “enough”.

---

> > ### Comment · Reviewer_xq3y · 2021-11-16
> > **Response**
> >
> > $\textbf{Matrix Decomposition}$
> >
> > Obviously, I know a diagonalizable matrix is eigendecomposable and symmetric is not a necessary condition. I believe the authors misunderstand my questions. I expect the authors to justify that the term $M^\top D_k(M-\gamma N)$ is eigendecomposable instead of referring to a common result and avoiding my question. Even the authors mention a diagonalizable assumption to make the matrix eigendecomposable, the assumption still needs to be justified.
> >
> > $\textbf{Not enough}$
> >
> > After referring to the recommended papers, I'm really happy to share my thoughts. I believe the authors should take more efforts to complete the paper by adding contents like asymptotic results and results with the wide 2-nn model. If the theoretical results are difficult to derive, I believe empirical studies are also sufficient.
> >
> > $\textbf{Effect of Thm 5}$
> >
> > As a reader, I don't under the effect of $d$ until seeing the authors' response. So I still highly recommend the authors give an explicit result in the main content. Besides, as $\lambda_{\min}\sim (\sqrt{d/n}-1)^2 $, it seems the first term in Corollary 2 would like to be a constant and the second term would be O(n). How do you explain an explosive upper bound on the generalization error?
> >
> > Oh I see. In over-parameterization regime, the dimension $d$ should be linear(or superlinear) in $n$. In that case, the generalization error is able to be controlled. I have no more puzzles in this part.
> >
> > $\textbf{Other questions}$
> >
> > 1. In section 3.1, the authors mention one can only expect to find an approximate solution that in general has nonzero Bellman error in the under-parameterized regime. I think that's also why many prior works consider the Completeness[1] assumption on the function hypothesis class to ensure efficient learning algorithms. I'd like to ask whether the Completeness assumption is no longer needed in the over-parameterized regime while zero-training loss is sufficient enough?
> >
> > 2. When it comes to theoretical comparison between over- and under- parameterization, I expect the authors could tell me what are the pros and cons when applying over-parameterization regime instead of under-parameterization. I believe a specific MDP example would be sufficient much more persuasive.
> >
> > 3. Besides, after roughly reading the recommended references [1,2,3], I'm much more confused. It seems the authors replace the linear regression setting with OPE without special phenomena in RL, especially [3]. [As I have insufficient time to make detailed reading on those references, I'm happy to listen to other reviewers' opinions. This point has no effect on my score judgment.]
> >
> > 4. Another error in Corollary 2. $n_{\min}$ is a random variable. I believe the authors should replace it with $n\min_s \mu(s)$ by applying Chernoff's inequality.
> >
> > $\textbf{Conclusion}$
> >
> > Overall, I believe the authors' response did better answer all my prior questions, but I don't feel it is enough to change my rating.
> >
> > [1] Jinglin Chen and Nan Jiang. Information-theoretic considerations in batch reinforcement learning. In
> > International Conference on Machine Learning (ICML), pages 1042–1051, 2019.

---

> > > ### Author Response · Authors · 2021-11-17
> > > **More Discussions (1/2)**
> > >
> > > $\def\mZ{\mathbf{Z}}$
> > > $\def\mM{\mathbf{M}}$
> > > $\def\mN{\mathbf{N}}$
> > > $\def\mQ{\mathbf{Q}}$
> > > $\def\mD{\mathbf{D}}$
> > > $\def\mJ{\mathbf{J}}$
> > > $\def\mI{\mathbf{I}}$
> > > $\def\mL{\mathbf{L}}$
> > >
> > > **Matrix Decomposition**
> > >
> > > The diagonalizable assumption is made for simplicity. In fact, we can use Jordan normal decomposition to show the general characterization.
> > >
> > > Let $\mZ= \mM^\top \mD_k (\mM-\gamma \mN)$ and $\mZ = \mQ \mJ \mQ^{-1}$ be the jordan normal form of $\mZ$.
> > > We still denote $\mQ_0$ the eigenvectors with eigenvalue zero.
> > > Then there is
> > > \begin{align}
> > >     \lim_{t\rightarrow \infty} (\mI - \eta \mZ)^t
> > >     =
> > >     \lim_{t\rightarrow \infty} \mQ (\mI - \eta \mJ)^t \mQ^{-1}
> > > \end{align}
> > > Since $\mI - \eta \mJ$ has a block diagonal structure, its power can be obtained by first computing the power of each block.
> > > Let $\mJ_i$ be the jordan block with eigenvalue $\lambda_i$.
> > > We write $\mJ_i = \lambda_i \mI + \mL$, where $\mL$ is a matrix such that the only non-zero entries of $\mL$ are on the first off-diagonal.
> > > Then we can write the $i$-th block of $\mJ$ as $(1-\eta \lambda_i)\mI - \eta \mL$.
> > > Using the binomial theorem we get
> > > \begin{align}
> > >     ((1-\eta \lambda_i)\mI - \eta \mL)^t =
> > >     \sum_{s=0}^t \tbinom{t}{s} (1-\eta \lambda_i)^{t-s} (-\eta \mL)^s\, .
> > > \end{align}
> > > Note that $\mL^s$ is the matrix with ones on the $s$-th diagonal away from the main diagonal, and $\mL^s=0$ for $s$ larger than the size of $\mL$.
> > > Therefore, $((1-\eta \lambda_i)\mI - \eta \mL)^t$ is a triangular matrix with $(1-\eta \lambda_i)^t$ on the main diagonal, $-\eta t (1-\eta \lambda_i)^{t-1}$ on the first off-diagonal, and so on.
> > > Therefore, the eigenvalues of this matrix are all $(1-\eta \lambda_i)^t$.
> > > Then given a learning rate that $\eta <  1/\lambda_{\max}$, for any jordan block with $\lambda_i>0$, we have that the matrix power converges.  For $\lambda_i = 0$, the jordan block corresponds to eigenvectors that are in the kernel space of $\mZ$.
> > > Thus, suppose that all eigenvalues of $\mZ$ are non-negative, we have
> > > \begin{align}
> > >     \lim_{t\rightarrow \infty} \mQ (\mI - \eta \mJ)^t \mQ^{-1} \theta_0
> > >     = \mQ_0 \mQ_0^{-1} \theta_0\,.
> > > \end{align}
> > > Note that if a negative $\lambda_i$ exists, the above derivations can still be used to characterize the convergent sub-component of $\theta_0$.
> > > The non-convergent sub-component of $\theta_0$ will diverge with an exponential rate as shown above.
> > >
> > > Again, we want to emphasize that this part has no effect on the main results of the paper. But we do thank the reviewer for the careful proof reading and spot this issue. Please let us know if the reviewer still has questions about this.
> > >
> > >
> > > **Note enough**
> > >
> > > As we have shown in the previous discussion, our result can be directly applied to non-linear function approximation using neural tangent kernel (NTK). By the neural tangent kernel definition (see the previous discussion), the neural tangent kernel corresponds to linear features $\phi(s) = \frac{\partial v(s,\omega)}{\partial \omega}$, where $\omega$ is the parameter of a neural network. The dimensionality of this linear feature $d$ is the number of parameters of the neural network. In NTK, $d>>n$, which is exactly the case this paper considered. One can just think that each row of $\mathbf{M}$ is $\phi(s)$ for the corresponding state $s$, and similarly for $\mathbf{N}$ and $\mathbf{M} - \gamma \mathbf{N}$. Please see the Appendix of the revision for an example of extending FVI convergence result using NTK.
> > >
> > > **Effect of Thm 5**
> > >
> > > Thanks for the suggestion, we agree that such result should be explicitly discussed in the paper.

---

> > > ### Author Response · Authors · 2021-11-17
> > > **More Discussions (2/2)**
> > >
> > > **I'd like to ask whether the Completeness assumption is no longer needed in the over-parameterized regime while zero-training loss is sufficient enough**
> > >
> > > The reason is that because of overparameterization (d>n), the Bellman constraints are actually an underdetermined linear system, which has  infinitely many feasible solutions.
> > >
> > >
> > > **I expect the authors could tell me what are the pros and cons when applying over-parameterization regime instead of under-parameterization**
> > >
> > > We never argue that there are pros and cons when applying overparameterization instead of underparameterization.
> > >
> > > The motivation of this paper is very clear. To better understand RL in the modern overparameterized regime, we must first understand the most basic overparameterized linear case. When studying value estimation in this case, the classical characterization of algorithm fixed points (MSPBE, MSTDE, MSBE…) collapse, as there is no approximation error and one can exactly drive Bellman errors to zero. Thus a natural question will be, do different algorithms (TD, RM, FVI..) still have different fixed points in the overparamterized setting? And it is indeed this very paper to give an answer to this question for the first time in the literature: the algorithms have certain implicit biases that become distinguishable in the overparameterized case.
> > >
> > > **Besides, after roughly reading the recommended references [1,2,3], I'm much more confused. It seems the authors replace the linear regression setting with OPE without special phenomena in RL, especially [3].**
> > >
> > > The results this paper presented are definitely not simply replacing the linear regression setting with OPE. Although one might consider RM as linear regression (by fitting reward using the residual features), it is definitely not true for TD and FVI. In fact, the learning dynamic of TD does not follow the gradient of any objective function (because of the semi-gradient update), and thus it is much more complicated to analyze than linear regression.
> > >
> > > Also, the generalization error bound can not be simply implied by the linear regression result in the previous papers. FVI can be thought as approximate dynamic programming using the current estimation at each iteration. One should explicitly compensate the approximation errors for each steps.
> > >
> > > **Another error in Corollary 2**
> > >
> > > $n_\min$ considers when the dataset if given and fixed for simplicity. But one could definitely consider the randomness of that also, which as the reviewer suggested just need an argument using Chernoff bound on random Bernoulli variable. We will add this in the paper. Thanks for the careful reading.

---

> ### Author Response · Authors · 2021-11-16
> **Discussion with Reviewer xq3y (2/2)**
>
> **Theoretical comparison between over- and under-parameterization**
>
> The comparison has explicitly been discussed in the paper. In the classical underparameterized setting, it is well known that TD solves the mean squared projected Bellman error (MSPBE), while RM solves the mean squared Bellman error (MSBE). However, these two solutions collapse when one can exactly achieve zero Bellman error for all training transitions, which suggests that the classical views fail in the overparameterized regime. Theorem 1, 2 and 3 in the paper shows that TD, FVI and RM do converge to different fixed points in the overparameterized linear case, which can be understood as a part of the implicit bias in supervised learning with SGD [4].
>
> **Effect of d in Theorem 5**
>
> The effect of $d$ can be observed though the smallest non-zero eigenvalue of the covariance matrix $\lambda_\min(\hat{\Sigma})$ in Corollary 2. For example, Bai-Yin limit shows that asymptotically $\lambda_\min\sim (\sqrt{d/n} - 1)^2$ [5]. We on purpose use the smallest eigenvalue in the bound to show the “essential dimension”. Note that the connection between $d$ and $\lambda_\min(\hat{\Sigma})$ has recently been used to explain the double descent phenomenon in deep learning [1,2,3].
>
>
> **References:**
>
>
> [1] Bartlett, Peter L., Andrea Montanari, and Alexander Rakhlin. "Deep learning: a statistical viewpoint." arXiv preprint arXiv:2103.09177 (2021).
>
> [2] Bartlett, Peter L., Philip M. Long, Gábor Lugosi, and Alexander Tsigler. "Benign overfitting in linear regression." Proceedings of the National Academy of Sciences 117, no. 48 (2020): 30063-30070.
>
> [3] Kuzborskij, Ilja, Csaba Szepesvári, Omar Rivasplata, Amal Rannen-Triki, and Razvan Pascanu. "On the role of optimization in double descent: A least squares study." arXiv preprint arXiv:2107.12685 (2021).
>
> [4] Soudry, Daniel, Elad Hoffer, Mor Shpigel Nacson, Suriya Gunasekar, and Nathan Srebro. "The implicit bias of gradient descent on separable data." The Journal of Machine Learning Research 19, no. 1 (2018): 2822-2878.
>
> [5] Bai, Zhi-Dong, and Yong-Qua Yin. "Limit of the smallest eigenvalue of a large dimensional sample covariance matrix." In Advances In Statistics, pp. 108-127. 2008.

---

> ### Author Response · Authors · 2021-11-24
> **Discussion**
>
> We are wondering if our responses addressed your concerns. We will be happy to answer if there are additional issues/questions.

---

> ### Author Response · Authors · 2021-12-02
> **Discussion**
>
> We are wondering if your question has been answered yet. Please let me know if you need more details.

---

### Official Review · Reviewer_KksW · 2021-11-02

**Correctness:** 3
**Technical Novelty And Significance:** 3
**Empirical Novelty And Significance:** 2
**Recommendation:** 6
**Confidence:** 2

**Main Review:**

### Strengths
Overall, this paper provides a novel perspective to reveal the implicit bias behind these classical value estimation algorithms, which is something that cannot be observed in under-parameterized case. The proposed regularizers are based on the weaknesses revealed by previous theoretical analysis and show good performance in experiments.

### Weaknesses
One of my major concerns is about the setting used in theoretical analysis. In particular, the analysis of this paper relies on an IID dataset $\left\lbrace (s_i, r_i, s_i')\right\rbrace_{i=1}^n$. However, in real offline policy evaluation problem, the dataset usually consists of sampled trajectories, in which the tuples $(s_i, r_i, s_i')$ are mutually dependent. It is not clear whether this dependency will break the results in this paper.

Another concern I have is about the experiments. Currently, the proposed regularizers are mainly tested in relatively simple environments. Is that possible to check whether these two regularizers are also effective in more complicated environments?

### Questions
- What if the dataset consists of many sampled trajectories instead of independent one-step transitions?
- What is the limit of generalization bound in Theorem 5 when $t\rightarrow\infty$ if the convergence criteria $||\mathbf{W}||\leq\frac{1}{\gamma}$ holds? How should we interpret this limit?

---

My concerns have been well-addressed and thus I increased my score. Meanwhile, I also decreased my confidence score because I realized that my expertise in this area might still need some improvement.

**Summary Of The Paper:**

This paper studies the convergence properties of three classical value estimation algorithms (TD, FVI and RM) under over-parameterized linear case. The difference among convergence results are interpreted unifiedly through different constraints in an optimization problem. It also proposes an generalization bound for FVI. Furthermore, based on the results mentioned before, it proposes two regularizers to help convergence in deep reinforcement learning and experimentally evaluates their performance.

**Summary Of The Review:**

This paper proposes a novel convergence analysis of three classical value estimation algorithms and reveals their implicit bias. However, the validity of its setting is questionable and the experiment results is not very persuasive.

---

> ### Author Response · Authors · 2021-11-16
> **Discussion with Reviewer KksW**
>
> We thank the reviewer for the work and input. The major concern of the reviewer is that the IID setting we considered is not interesting and our contributions are limited. We think this is unfounded and explain why below.
>
> 1. The IID data assumption is the most widely used assumption in offline reinforcement learning, for example, see [1,2,3,4,5,6,7].
>
> 2. Given that the MDP under the behavior policy that is used to collect the data is ergodic, the stationary occupancy measure $\mu$ must exist. Thus, a dataset consists of many sampled trajectories can also be thought of as generating data by $s\sim \mu, r = r(s), s’\sim P(\cdot | s)$.
>
> 3. We note that the convergence results (Theorem 1, 2 and 3) and the unified view (Theorem 4 and Corollary 1) do not use the IID data assumption. All results can be easily generalized to the classical settings where one considers the fixed points under expected update [8,9,10,11].
>
> 4. The reason why we consider the batch IID setting is that it provides a clean setup to study the value estimation algorithms in an over parameterized linear regime. And in fact, it is actually good enough to show some critical observations that cannot be simply implied by the results in the under parameterized regime. Just like one does not build a house starting with the roof, we maintain that one should start with asking the simplest questions before adding “bells and whistles”.
>
>
> Other questions:
>
> **What if the dataset consists of many sampled trajectories?**
>
> The convergence results (Theorem 1, 2 and 3), and the unified view (Theorem 4 and Corollary 1) do not use the IID data assumption. For the batch setting, even with the trajectory-based samples, the generalization error can be directly applied by making certain coverage assumptions on the stationary distribution $\mu$ (see point 2 in the response above), see [12] for example.
>
> **Experiments**
>
> We emphasize that the main contributions are theoretical understandings for value estimation algorithms in the overparameterized linear regime. The experiments are just designed to justify the proposed regularizations inspired by the theoretical findings. We add new experiment results on four Mujoco control problems in the Appendix B.5 of the revision.
>
> **What is the limit of generalization bound in Theorem 5 when $t\rightarrow \infty$ if the convergence criteria holds**
>
> This is exactly the result shown in Corollary 2, where the value prediction error is decomposed into the variance (first term) and the bias (second term). The variance is due to using finite samples, while the bias quantifies the orthogonal component of the optimal weight vector with respect to the spectrum of features (i.e. the approximation error).
>
> Theorem 5 gives the value generalization error bound of FVI **per iteration**. When $||W||<1/\gamma$, FVI will converge to the exactly same solution as TD. That means, Corollary 2 also gives the value generalization error of TD fixed point under the convergence condition.
> When the convergence condition does not hold, the value prediction error of FVI and TD fixed point can be arbitrarily large. This also connects to a similar result for the TD fixed point that minimizes MSPBE in the underparameterized regime [9].

---

> ### Author Response · Authors · 2021-11-16
> **References**
>
> [1] Ofir Nachum, Yinlam Chow, Bo Dai, and Lihong Li. "Dualdice: Behavior-agnostic estimation of discounted stationary distribution corrections." arXiv preprint arXiv:1906.04733 (2019).
>
> [2] Aviral Kumar, Aurick Zhou, George Tucker, and Sergey Levine. Conservative q-learning for oine reinforcement learning. In NeurIPS, volume 33, pages 1179–1191, 2020.
>
> [3] Ming Yin, Yu Bai, and Yu-Xiang Wang. Near-optimal provable uniform convergence in online policy evaluation for reinforcement learning. In AISTATS, pages 1567–1575, 2021
>
> [4] Ruosong Wang, Dean P. Foster, and Sham M. Kakade. What are the statistical limits of online RL with linear function approximation? In ICLR, 2021.
>
> [5] Masatoshi Uehara, Masaaki Imaizumi, Nan Jiang, Nathan Kallus, Wen Sun, and Tengyang Xie. Finite sample analysis of minimax online reinforcement learning: Completeness, fast rates and first-order efficiency. arXiv preprint arXiv:2102.02981, 2021.
>
> [6] Yao Liu, Adith Swaminathan, Alekh Agarwal, and Emma Brunskill. Provably good batch o-policy reinforcement learning without great exploration. In NeurIPS, 2020.
>
> [7] Levine, S., Kumar, A., Tucker, G., & Fu, J. (2020). Offline reinforcement learning: Tutorial, review, and perspectives on open problems. arXiv preprint arXiv:2005.01643.
>
> [8] Tsitsiklis, John N., and Benjamin Van Roy. "An analysis of temporal-difference learning with function approximation." IEEE transactions on automatic control 42, no. 5 (1997): 674-690.
>
> [9] Kolter, J. "The fixed points of off-policy TD." Advances in Neural Information Processing Systems 24 (2011): 2169-2177.
>
> [10] Dann, Christoph, Gerhard Neumann, and Jan Peters. "Policy evaluation with temporal differences: A survey and comparison." Journal of Machine Learning Research 15 (2014): 809-883.
>
> [11] Bhandari, Jalaj, Daniel Russo, and Raghav Singal. "A finite time analysis of temporal difference learning with linear function approximation." In Conference on learning theory, pp. 1691-1692. PMLR, 2018.
>
> [12] Jin, Y., Yang, Z. and Wang, Z., 2021, July. Is Pessimism Provably Efficient for Offline RL?. In International Conference on Machine Learning (pp. 5084-5096). PMLR.

---

> ### Author Response · Authors · 2021-11-24
> **Discussion**
>
> We are wondering if our responses addressed your concerns. We will be happy to answer if there are additional issues/questions.

---

> > ### Comment · Reviewer_KksW · 2021-11-24
> > **Response**
> >
> > Thank you very much for your response. My concerns have been well-addressed and thus I increased my score. Meanwhile, I'm sorry for being unprofessional in part of my reviews and thus I also decreased my confidence score.

---

> > > ### Author Response · Authors · 2021-11-25
> > > **Response**
> > >
> > > Thank you for your time reviewing our response. We are glad to hear that your concerns have been addressed.

---

### Official Review · Reviewer_PSbG · 2021-11-04

**Correctness:** 4
**Technical Novelty And Significance:** 3
**Empirical Novelty And Significance:** 1
**Recommendation:** 8
**Confidence:** 4

**Main Review:**

In the reinforcement learning literature, numerous algorithms exist to learn the value function based on bootstrapping and in presence of function approximation. Chief among these algorithms are (linear) temporal difference learning and the residual gradient. These two differ slightly in terms of the objective functions (MSPBE vs MSBE), but it is known that this small difference can manifest itself significantly in practice. The commonly held intuition is that the TD algorithm has a superior fixed point, so eventhough Maei showed that TD is not following the gradient of any objective, it is preferred to residual gradient in practice. The soundness of residual gradient is also overshadowed by its need for double sampling to combat the bias that would otherwise be present.

One may intuit that the difference between the fixed point of the two approaches may not carry over to the setting with sufficiently rich feature space, which is the central question of this paper. The paper shows that indeed even in the overparameterized setting, the two approaches may result in different fixed-points. More specifically, each approach can have numerous fixed-points and convergence to a particular fixed point depends upon the initial weight vector, but that crucially these fixed points are different on a per-algorithm basis. This makes sense. After all, the parameter vector is trained to predict different things in each case: value function in TD, and temporal differences in residual gradient.

Overall, reading this paper really excited me, eventhough sometimes it felt like that the paper is purposefully making the math to look more difficult than it needs to be. It is certainly very interesting to study the fixed point of different approaches in settings that more closely resemble the common practice in which people use extremely powerful function approximators.

Getting to the core of the results, I have noticed a few issues worth mentioning:

- The fixed point in theorem 2 is in fact extremely similar to equation (2) from [1]. The newly introduced variables make it less obvious to see, but after plugging in the relevant quantities, one can see that things cancel out and lead to a solution very similar to what i noted, and the paper itself does note in equation (7). This result is still novel and does generalize to the underdetermined case, but I don't feel great about the paper not giving enough credit to [1] in the context of this result.

- The result branded as theorem 4 is really trivial. In particular, this is nothing but the least norm solution for underdetermined equations. See for example: https://see.stanford.edu/materials/lsoeldsee263/08-min-norm.pdf. Again, branding this as a theorem and not giving credit to people who first came up with this result is not a great practice.

- What's even more surprising, the Proof of theorem 4 in the appendix, which is just a copy of the existing result from the convex optimization literature, includes several mistakes. For example, as written $L(\theta, \alpha)$ is not a function of $\theta$ nor $\alpha$! It is in fact constant with respect to these variables. I think authors meant to compute $\inf_\theta \sup_\alpha L(\theta,\alpha)$ and refer to the objective as $L(\theta,\alpha)$. In any case, as written, this gives the wrong impression that this result is first derived in this work, but this is just a trivial result in convex optimization as I mentioned above.

- Less severe issue, but In equation 38 in the Appendix, it seems like that t+1 should actually be $\infty$?

- While I think the studied setting is very interesting, I don't quite see the connection with deep RL. In deep RL, it is not that the number of features are extremely large, therefore the last layer is solving an underdetermined problem. It is more that all but the last layer are jointly learning a basis function given which it is very easy to fit the value function using linear function approximation. For example, the last layer of DQN is comprised of 256 features, yet on a given game the support of the stationary distribution is vastly larger than 256 in light of the combinatorial nature of most games, how do you square your relating the setting to deep RL with my observation?

- I am also not sure I learned a lot from the experimental sections. Domains are pretty toyish and experiments are not well motivated. Also, the figures and their captions look pretty small. In light of the theory it is OK to not have a significant empirical contribution, but at this point, its debatable if the experiments are adding something or actually degrading the paper.



[1] Parr and friends, "An Analysis of Linear Models, Linear Value-Function Approximation, and Feature Selection for Reinforcement Learning"


**Summary Of The Paper:**

This paper studies the fixed point solution of standard model-free value-function optimization algorithms in RL (in particular TD, residual gradient, and Fitted Value Iteration) in the over-parameterized setting, meaning that the problem is underdetermined and can generally admit numerous optimal solutions. Importantly the paper shows that the different fixed points of TD and residual gradient in the limited-capacity case carries over to the over-parameterized setting as well.

**Summary Of The Review:**

As mentioned above, I am excited about this paper in light of the importance of studying TD and related algorithms in settings other than standard under-parameterized settings. The demonstrated scholarship, or lack thereof, gives me doubts, but overall I like this paper and lean towards the acceptance side.
--
post rebuttal: I think this paper makes a respectable contribution to the theory of temporal difference learning. I believe authors will address my concerns, so I like to see this paper accepted.

---

> ### Author Response · Authors · 2021-11-16
> **Discussion with Reviewer PSbG**
>
> We thank the reviewer for acknowledging our contributions and providing insightful suggestions. The detailed responses are provided below.
>
> **The fixed point in theorem 2 is in fact extremely similar to equation (2) from [1].**
>
> We are quite familiar with and recognize the contribution of Parr et al., 2008, and indeed cite that paper in the related work (Section 2). Note that Eq (2) in Parr et al., 2008 is the TD fixed point in the under parameterized setting. We provide the solution in Eq (7) simply as part of the background, not the main contribution, while discussing the related works that study this fixed point in both Section 3.1 and Section 2.
>
>
> **The result branded as theorem 4 is really trivial.**
>
> We disagree with this claim. The major argument of Theorem 4 is to characterize the fixed points of TD and RM via optimization. In fact, neither RM nor TD **explicitly** introduces regularization in the algorithms, while the theorem reveals such implicit regularization effects by analyzing the convergent solution, which can be understood as a part of the implicit bias in supervised learning with SGD in [1].
>
> Furthermore, it is actually not trivial to discover such characterization. One must first know the exact form TD fixed point, and realize that the solution is within the feature space instead of the residual features as RM, which is indeed the contribution of this very paper.
>
>
> **as written $L(\theta, \alpha)$ is not a function of $\theta$ nor $\alpha$! It is in fact constant with respect to these variables…Less severe issue, but In equation 38 in the Appendix…**
>
> Thanks for pointing out the typos. We have fixed these in the revision.
>
> **While I think the studied setting is very interesting, I don't quite see the connection with deep RL…**
>
> The connection between an overparameterized linear model and deep neural network can be established using neural tangent kernels. In Appendix A.8 of the revision, we show how to derive the convergence of FVI using the neural tangent kernel as an example. Here we briefly discuss the idea.
>
> Let $v(s, \omega)$ be the value function approximated by a neural network, where $\omega$ is all the parameters of the neural network.
> The neural tangent kernel is defined as
> \begin{align}
>     \text{ker}(s, s') =
>     \mathbb{E}_{\omega\sim\Omega}
>     \left\langle
>     \frac{\partial v(s, \omega)}{\partial \omega},
>     \frac{\partial v(s', \omega)}{\partial \omega}
>     \right\rangle
> \end{align}
> where $s, s'$ are two states, $\Omega$ is an initialization distribution over $\omega$.
> Let $\omega_0$ be random initial parameters sampled from $\Omega$, there is
> \begin{align}
>     \text{ker}_0(s,s') = \left\langle
>     \frac{\partial v(s, \omega_0)}{\partial \omega_0},
>     \frac{\partial v(s', \omega_0)}{\partial \omega_0}
>     \right\rangle
>     \approx
>     \text{ker}(s, s')\, .
> \end{align}
> This approximation is accurate when the width of the neural network is allowed to increase to infinity.
> See Theorem 3.1 in [2] for the proof. Note that in this kernel, an input $s$ is mapped to a feature vector $\phi(s) = \frac{\partial v(s, \omega_0)}{\partial \omega_0}$, which is a $d$-dimensional linear feature where $d$ is the total number of parameters of the neural network. The main results of this paper can be extended to this setting using the kernel trick.
>
>
> **I am also not sure I learned a lot from the experimental sections. Domains are pretty toyish and experiments are not well motivated.**
>
>
> We agree that the main contributions are theoretical understandings. The experiments are just designed to justify the proposed regularizations inspired by the theoretical findings. We add new experiment results on four Mujoco control problems in the Appendix B.5 of the revision.
>
>
> [1] Soudry, Daniel, Elad Hoffer, Mor Shpigel Nacson, Suriya Gunasekar, and Nathan Srebro. "The implicit bias of gradient descent on separable data." The Journal of Machine Learning Research 19, no. 1 (2018): 2822-2878.
>
> [2] Arora, Sanjeev, Simon S. Du, Wei Hu, Zhiyuan Li, Ruslan Salakhutdinov, and Ruosong Wang. "On exact computation with an infinitely wide neural net." arXiv preprint arXiv:1904.11955 (2019).

---

> > ### Comment · Reviewer_PSbG · 2021-11-30
> > **Response**
> >
> > Sorry for the delay, but I am not sure what you are disagreeing with. I am simply saying, the solution to (71) is already known. Branding it as a proof without proper recognition of its knownness seems weird to me.
> >
> > The connection with infinite-width neural nets is definitely interesting, but that does not address my concern, right? That in particular, the width of the nets used in deep RL is often not that large.

---

> > > ### Author Response · Authors · 2021-12-01
> > > **Response**
> > >
> > > Thanks for your response.
> > >
> > > **I am simply saying, the solution to (71) is already known. Branding it as a proof without proper recognition of its knownness seems weird to me.**
> > >
> > > We thought the reviewer is challenging the novelty of the unified view. But it seems that the reviewer was only suggesting to add a reference for that solution. We completely agree with that. Sorry for the confusion.
> > >
> > > **The connection with infinite-width neural nets is definitely interesting, but that does not address my concern, right?**
> > >
> > > Yes we agree that one can interpret the value network prediction as a linear function by considering the outputs of the penultimate layer as features, and the width of the net used in deep RL is often not that large. But, that does not mean it is exactly the same as the under parameterized case, since the neural network has power to learn the feature to exactly fit all the data. We think it is actually an open problem that how to interpret a neural network as a linear function of some basis. Neural tangent kernel is a very interesting theory that tries to make such connection, and from the NTK perspective, the linear features are actually the gradient of the parameters, which is definitely over parameterized. But we agree the reviewer definitely makes a very good point, and we will provide some discussion about this point in the paper. We would also like to discuss this further with the reviewer.

---

> > > > ### Comment · Reviewer_PSbG · 2021-12-01
> > > > **Response**
> > > >
> > > > Great. I am increasing my score. Please make sure 1) the paper gives more credit to existing results whenever its due and 2) the paper is not making strong claims about connections to deep RL.

---

> > > > > ### Author Response · Authors · 2021-12-02
> > > > > **Response**
> > > > >
> > > > > We are glad that the reviewer's concerns have been addressed. Thanks for the valuable feedback and a positive assessment of our work.

---

> ### Author Response · Authors · 2021-11-24
> **Discussion**
>
> We are wondering if our responses addressed your concerns. We will be happy to answer if there are additional issues/questions.

---

### Decision · Program_Chairs · 2022-01-20

**Decision:**

Accept (Poster)

**Comment:**

This paper presents a study of the over parametrization of linear representations in the context of recursive value estimation.

The reviewers could not reach a consensus over the quality of the paper, with a fairly wide range of scores even after the rebuttal.

After considering the paper, the rebuttal, and the discussion, I lean towards accepting the paper. Despite the concerns voiced by some of the reviewers, the topic and analysis of the manuscript are novel and interesting, and it is my expectation that this manuscript will prove a valuable source of inspiration for future work.

I invite the authors to carefully consider the feedback received by all the reviewers (and in particular Reviewers xq3y and gT5o and) and to revisit the manuscript accordingly.